# Intra-pixel variability in satellite tropospheric $NO_2$ column densities derived from simultaneous space-borne and airborne observations over the South African Highveld

Stephen Broccardo[1], Klaus-Peter Heue[2], David Walter[3], Christian Meyer[4], Alexander Kokhanovsky[5,6], Ronald v.d. A[7], Stuart Piketh[8], Kristy Langerman[9], and Ulrich Platt[10]

[1]School of Geography, Archaeology and Environmental Science, University of the Witwatersrand, Johannesburg, 2030, South Africa, *now at* Climatology Research Group, Unit for Environmental Science and Management, North-West University, Potchefstroom, 2531, South Africa
[2]DLR Earth Observation Center, Oberpfaffenhofen, 82234 Wessling, Germany
[3]Max Planck Institut für Chemie, Hahn-Meitner-Weg 1, 55128 Mainz, Germany
[4]IDT Europe GmbH, Grentzstr 28, 01109 Dresden, Germany
[5]EUMETSAT, Eumetsat Allee 1, 64295, Darmstadt, Germany
[6]Moscow Engineering Physics Institute, National Research Nuclear University, Kashirskoe Ave. 31, 115409, Moscow, Russia
[7]R&D Satellite Observations, KNMI, Utrechtseweg 297, 3731GA, De Bilt, Netherlands
[8]Climatology Research Group, Unit for Environmental Science and Management, North-West University, Potchefstroom, 2531, South Africa
[9]Eskom Holdings SOC Ltd, Megawatt Park, Maxwell Drive, Sandton, 2157, South Africa
[10]Institut für Umweltphysik, Im Neuenheimer Feld 229, 69120, Heidelberg, Germany

*Correspondence to:* S. Broccardo (sbroccardo@gmail.com)

**Abstract.** Aircraft measurements of $NO_2$ using an imaging differential optical absorption spectrometer (iDOAS) instrument over the South African Highveld region in August 2007 are presented and compared to satellite measurements from OMI and SCIAMACHY. In-situ aerosol and trace-gas vertical profile measurements, along with aerosol optical thickness and single-scattering albedo measurements from the Aerosol Robotic Network (AERONET), are used to devise scenarios for a radiative-transfer modelling sensitivity study. Uncertainty in the air-mass factor due to variations in the aerosol and $NO_2$ profile shape is constrained, and used to calculate vertical column densities (VCD), which are compared to co-located satellite measurements. The lower spatial resolution of the satellites cannot resolve the detailed plume structures revealed in the aircraft measurements. The airborne DOAS in general measured steeper horizontal gradients and higher peak $NO_2$ vertical column density. Aircraft measurements close to major sources, spatially-averaged to the satellite resolution, indicate $NO_2$ column densities more than twice those measured by the satellite. The agreement between the high-resolution aircraft instrument and the satellite instrument improves with distance from the source, this is attributed to horizontal and vertical dispersion of $NO_2$ in the boundary layer. Despite the low spatial resolution, satellite images reveal point sources and plumes that retain their structure for several hundred kilometers downwind.

# 1 Introduction

Space-based measurements of trace-gases are increasingly being used to monitor tropospheric air pollution (McLinden et al., 2012; Streets et al., 2013), including the identification of major sources missing from public emissions inventories (Mclinden et al., 2016) and the quantification of source strengths (Beirle et al., 2011). Satellite observations have highlighted the South African Highveld as a region with $NO_2$ column densities higher than expected from emissions inventories (Martin et al., 2002; Toenges-Schuller et 2006), and with an increasing trend (Richter et al., 2005; van der A et al., 2008). To further investigate this phenomenon, a high-resolution imaging differential optical absorption spectrometer (iDOAS) was flown on board the South African Weather Service Aerocommander 690A research aircraft during a dry-season flight campaign over the Highveld in 2007. Results from research flights conducted on the 9th, 11th, 14th and 18th of August 2007 are presented. Aircraft vertical profile measurements of $NO_y$ and aerosols are used to devise several representative scenarios of the vertical distribution of these two species, and a sensitivity study is performed using the SCIATRAN radiative transfer model (Rozanov et al., 2014) to constrain the uncertainty in the air-mass factor. Measurements from the nadir pixel of the iDOAS are compared with operational satellite measurements of $NO_2$ from OMI (Ozone Monitoring Instrument) and SCIAMACHY (SCanning Imaging Absorption spectroMeter for Atmospheric CHartographY) made on the same day.

The Highveld is a high-altitude plateau in the interior of South Africa (Fig. 1), home to the Johannesburg-Pretoria conurbation and the adjoining industrial towns of Ekurhuleni to the east and Vereeniging and Vanderbijlpark to the south. The latter two along with the petrochemical industry and town at Sasolburg enclose an area known as the Vaal Triangle. The Vaal Triangle is also home to two steel-mills and a coal-fired power station. To the east of Johannesburg, at Secunda, there is a coal-to-fuel (Fischer-Tropsch process) synfuel refinery, which also generates electricity from coal. Secunda is situated in a region often referred to as the Eastern Highveld or Mpumalanga Highveld; this region is also home to eleven more coal-fired power stations, and several steel-mills. Analysis of a year of ground-based monitoring station data by Collett et al. (2010) indicates that most of the $NO_2$ on the Highveld is from tall-stack industrial emissions, and has an impact on surface ozone concentrations (Balashov et al., 2014). A combined analysis of satellite and ground-based measurements indicates that the conurbation of greater Johannesburg is also a significant source of $NO_2$ (Lourens et al., 2012). In between the heavy industries are coal mines to supply fuel to the former, a small town about every hundred kilometers, and farmland or grassland. The Highveld is impacted by biomass-burning sources in the winter season; along with the urban and industrial sources of trace-gases and aerosols, it forms a natural laboratory isolated on a global scale from nearby sources, and controlled by synoptic-scale meteorology (Annegarn et al., 2002).

# 2 Measurements and Methods

## 2.1 DOAS

The measurement principle employed to observe $NO_2$ from satellite, and from our airborne iDOAS instrument is that of differential optical absorption spectroscopy (DOAS), described by Platt and Stutz (2008). As with many absorption spectroscopy

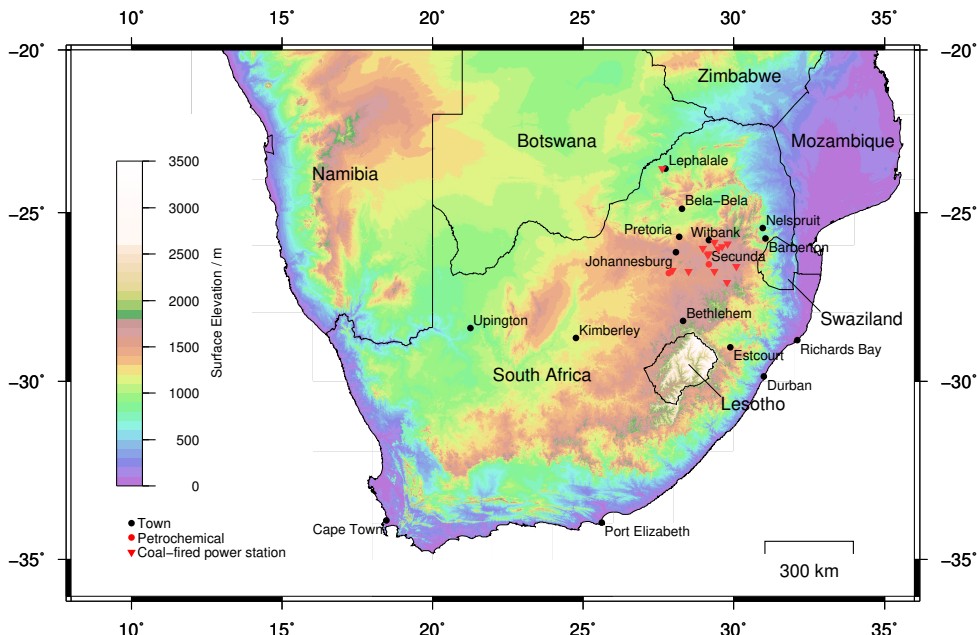

**Figure 1.** A map of Southern Africa, showing the high-altitude plateau of the Highveld to the east of Johannesburg, and the cluster of coal-fired power stations and heavy industries on the eastern Highveld. The coal-fired power station at Lephalale is indicated. Power stations not shown here include smaller coal-fired and gas-turbine power stations operated by the cities.

techniques, the magnitude of the measured quantity depends on the path length, $p$, through the absorber and the molecular number density, $n(p)$, of the absorber. In the case of measurements made in the atmosphere using scattered sunlight, the DOAS analysis yields a slant column density (SCD), $S$:

$$S = \int_{path} n(p)dp. \tag{1}$$

5    This name reflects the fact that the light path through the atmosphere is not known *a-priori*, and to a first geometric approximation, is slanted. A more useful quantity is the vertical column density (VCD), which is the molecular number density of the absorber integrated along a vertical path between the earth's surface and the top of the atmosphere. In the standard DOAS formulation suited to measurement of relatively small slant-column densities (Rozanov and Rozanov, 2010), these two quantities are related by an air-mass factor (AMF):

10   $$A = S/V \tag{2}$$

where $A$ is the air-mass factor, $V$ is the vertical column density and $S$ is the slant column density as before.

A DOAS instrument based on an Acton 300i imaging spectrograph employing a pushbroom viewing geometry, where each line of pixels across the instrument's swath is captured simultaneously on an Andor DU-420BU CCD camera, was fitted into the Aerocommander 690A research aircraft. This CCD has 255 pixels in the across-track dimension and 1024 pixels in the

spectral direction. The temperature of the spectrograph was kept stable at 30°C using a thermostatic heater in an insulated box, and the CCD temperature was set at -20°C using its own in-built thermo-electric cooler. Eight spectra were co-added into 32 across-track pixels, each with an across-track footprint of approximately 70-80m, assuming a flight altitude of 4500m above the ground. This was done in order to make optimum use of the optical resolution of the instrument. Along-track resolution is

determined by the aircraft speed and the integration time of the instrument, which was adjusted automatically in-flight to avoid saturation of the CCD, but is generally about 100m (Heue et al., 2008). In the present study only the nadir pixel of the iDOAS is used.

Slant-column densities were retrieved using the WinDOAS software package. Absorption cross-sections for $NO_2$ (Vandaele et al., 1998), ozone (Burrows et al., 1999), water vapour (Rothman et al., 1998), $O_4$ (Greenblatt et al., 1990) were fitted across a

spectral range of 432nm to 464nm. The Ring effect was accounted for using a appropriate cross-section calculated using the DOASIS software (Kraus, 2006). A reference spectrum was chosen from an appropriate location along the flight track far from known sources implying that slant-column densities from WinDOAS are in fact differential slant column densities. Satellite retrievals use a similar technique, using a measurement over remote ocean areas as an approximation of zero-$NO_2$. We adjust our slant-column densities using an offset in order to bring the vertical column densities from the iDOAS into line with the

appropriate satellite measurement (either OMI or SCIAMACHY) in background areas of our flight track.

Two approaches are taken in order to allow a comparison of the iDOAS with the satellite-based measurements: the first is to average 80m-resolution nadir iDOAS measurements using a ten-second moving average in order to smooth out fine-spatial-scale variations and make a comparison with the much larger satellite pixels. With the aircraft's ground speed being around $120 \mathrm{m\,s^{-1}}$, on a spatial scale this time-based moving-average is over approximately 1.2km. The second approach is to calculate

the mean and standard deviation of all nadir (80m by 100m) iDOAS measurements along the aircraft track within a satellite ground pixel to compare with the value from the satellite tropospheric $NO_2$ product for that pixel. This is referred to as a line-average.

## 2.2   In-situ measurements

In addition to the imaging DOAS (iDOAS) instrument, the aircraft carried a Particle Measurement Systems Passive Cavity

Aerosol Spectrometer Probe 100X (PCASP), operated with the pre-heater switched on; and a Thermo Scientific 42i chemi-luminescence instrument with a molybdenum converter in the cabin, plumbed into the aircraft's scientific-air inlet in order to measure in-situ $NO_y$. In such instruments the converter converts $NO_2$ to NO, which is then measured by chemiluminescence; however a molybdenum converter also converts other nitrogen species which comprise $NO_y$. This can be avoided using a photolytic converter, however an instrument with a photolytic converter to measure $NO_2$ was not within the project's budget. The

aircraft is fitted with a Rosemount ambient temperature sensor, and a separate pitot-static system for measurement and logging of static and dynamic pressure. The humidity sensor fitted to the aircraft did not function during this campaign. The aircraft's data acquisition system also logged parameters from a GPS (Global Positioning System) receiver.

Aerosol number concentration and in-situ $NO_y$ are averaged into 50m altitude bins, temperature into 20m bins. Altitude intervals of interest are identified by inspection of the vertical profile measurements, and average particle size spectra are calcu-

lated. No corrections for aerosol refractive index are made to the PCASP measurements (Rosenberg et al., 2012; Liu and Daum, 2000), since the present measurements are not used for determination of radiative properties of the aerosols.

## 2.3 Satellite measurements

Satellite-based measurements of $NO_2$ were made operationally from the SCIAMACHY (Scanning Imaging Absorption spectroMeter for Atmospheric CHartographY) instrument on board the European Space Agency (ESA) ENVISAT satellite from March 2002 to April 2012; and from OMI (Ozone Monitoring Instrument) on the National Aeronautics and Space Agency (NASA) Aura satellite from October 2004 until the present. The SCIAMACHY instrument operated in a whiskbroom geometry (where the instrument's field-of-view is scanned from side-to-side across the swath) with eight measurement channels covering the spectral range from 214nm to 2386nm. ENVISAT orbited at a mean altitude of 799.8km with an orbital period of 100.6min and a repeat cycle of 35 days. Overpass time on the Highveld was around 10:00 local time. In the nadir viewing geometry the ground pixel size is 60km by 30km, and global coverage was achieved every 6 days (Gottwald et al., 2006). The Aura satellite orbits at a mean altitude of 709km, with an orbital period of 98.8min and a repeat cycle of 16 days. The OMI instrument measures wavelengths between 270nm and 500nm in pushbroom geometry with a nadir ground-pixel size of 24km by 13km (Levelt et al., 2006). OMI pixels broaden in the across track direction as the viewing angle moves away from nadir. Overpass on the Highveld is around 14:00 local time.

Measurements from the nadir pixel of the iDOAS are compared with $NO_2$ tropospheric VCD from the DOMINO (Derivation of OMI tropospheric $NO_2$) version 2.0 product from the OMI instrument (Boersma et al., 2011) available from http://www.temis.nl. *A-priori* vertical profiles of $NO_2$ from the TM4 global chemistry-transport model (Dentener et al., 2003) are used to calculate tropospheric air-mass factors, and stratospheric $NO_2$ is estimated by assimilation of slant columns in the TM4 model. The TM4NO2A product (also available from http://www.temis.nl) uses slant column measurements from the SCIAMACHY satellite instrument and a similar scheme using model profiles and stratospheric columns from the TM4 model.

## 2.4 Flight strategy

Flights were planned to approximately follow the nadir track of the satellite (carrying OMI or SCIAMACHY) that would be passing over the Highveld on the day, with the aircraft flying nominally at 6000m above sea level, the actual altitude varying by 1000ft (312m) as demanded by air-traffic rules. Over much of the Highveld, this would be approximately 4500m above the ground, giving ground pixels from the iDOAS approximately 80m by 100m (Heue et al., 2008).

At the beginning and end of the satellite-tracking segment of each flight, a vertical profile measurement of $NO_y$ and aerosols was performed. Vertical profiles were started and ended as low as safety allowed, judged visually to be around 400m–500m above ground level (AGL). The lower altitude limit of vertical profile measurements could be safely extended down to the surface if the profiles were flown overhead a suitable airfield, and the pilot performed a missed-approach procedure. This would limit the choice of locations for vertical profile measurements, but the quality of the profiles would be improved. Intermittent

failures of the PCASP probe and the data acquisition system detract from the usefulness of some of the profiles, and these partial profiles are not presented here.

## 2.5 AERONET measurements

Monthly statistics are calculated for the late-winter season from measurements of aerosol optical thickness (AOT) and single-scattering albedo (SSA or $\bar{\omega}$) taken from an AERONET sun-photometer (Holben et al., 2001) that was situated at the University of the Witwatersrand in Johannesburg during 2007 and 2009. The aircraft vertical profile measurements and sun-photometer measurements are used as guidance in creating a number of vertical profile scenarios of aerosols and $NO_2$; which are used in a model sensitivity study using the SCIATRAN radiative transfer model (RTM) (Rozanov et al., 2014). The results of this sensitivity study are used to constrain the uncertainty in the air-mass factor for the iDOAS $NO_2$ measurements.

## 3 Aircraft Vertical Profile Measurements of in-situ NOy and aerosols

During each of the flights, a vertical profile measurement was performed before and after the satellite-tracking portion of the flight. From the measured vertical profiles, several features can be discerned:

1. $NO_y$ and aerosol concentration profile shapes are block-shaped or exponentially-decreasing with altitude.

Since aerosol particles or their precursors, and $NO_y$, often are emitted from the same surface urban, industrial or biomass-burning sources, the patterns of their dispersal will be similar. A block-shaped vertical profile can be expected under conditions where turbulent mixing causes vertical dispersion in the planetary boundary layer, an exponentially-decreasing profile will occur under conditions of greater atmospheric stability, or close to sources where dispersion has not had an opportunity to take place.

2. There are elevated layers of enhanced aerosol and $NO_y$ concentration, isolated from the planetary boundary layer by a layer of cleaner air.

Swap and Tyson (1999) assess vertical mixing and transport of air parcels between spatially and temporally persistent stable layers over the sub-continent. These stable layers around 850hPa (in coastal areas), 700hPa, 500hPa, and 300hPa (corresponding roughly to 1500m, 3000m, 5800m and 9200m respectively) lead to peaks and discontinuities in the vertical profile of trace-gases and aerosols. Published vertical profile measurements of aerosol scattering coefficient (Magi et al., 2003) and particle concentration (Hobbs, 2003; Swap and Tyson, 1999) from the sub-continent show features consistent with this generalization.

3. The aerosol size distribution is consistent in the lower and upper sections the profile

Since the aerosols in the elevated layers are transported there from the lower layers (Swap and Tyson, 1999; Hobbs, 2003) where they are emitted or formed, the size distributions can be expected to be similar. This similarity in aerosol size distribution

through the vertical profile was also found in measurements over Namibia (Haywood, 2003a, b), a region frequently under the influence of the same sub-continental-scale air transport regime as the Highveld (Swap and Tyson, 1999). Assuming that aerosol optical properties are the same in the elevated layers as they are near the surface, similar size distributions mean that the aerosol scattering and absorption coefficients will be proportional to aerosol number concentration. These generalizations of the vertical profile are used to develop scenarios for a radiative-transfer modelling sensitivity study described below.

As examples of aircraft vertical profile measurements, the profile overhead the coastal town of Richards Bay measured on 11 August is shown in Fig. 2 and the profile overhead Nelspruit on the same day is shown in Fig. 3. The lower limit of the former profile was around 500m, the latter profile was measured down to the surface, since the aircraft landed at Nelspruit. Extrapolating the available measurements to the surface at Richard's Bay, the aerosol number and $NO_y$ concentrations appear to follow a generally exponentially-decreasing profile with height. The top of this exponential profile is at the bottom of a temperature inversion around 1750m (approximately 815hPa). Embedded within this profile there is a layer of elevated $NO_y$ between 700m and 1000m; this layer of enhanced $NO_y$ concentration is approximately mirrored in the aerosol profile. In addition to the exponentially-decreasing profile close to the ground there is a separate elevated layer of enhanced aerosol and $NO_y$, between 2200m and 2700m above sea level, capped by another temperature inversion at approximately 2700m (730hPa). Layer-averaged aerosol size spectra from the altitude intervals 500m–600m, 700m–900m, 1000m–1600m, and 2200m–2700m indicate a similarly-shaped bimodal log-normal distribution in all cases, with modes at 0.13µm and 2.25µm.

Over Nelspruit (Fig. 3), the aircraft descended through a plume between 2900m–2500m, observed to originate from a large forest-fire nearby; $NO_y$ concentrations up to 35ppb and aerosol number concentrations greater than 6000cm$^{-3}$ were measured in this plume. This large plume was trapped under a temperature inversion at 2900m AMSL (approximately 710hPa). Below this, between 2400m–2100m a layer of enhanced aerosol number concentration, and $NO_y$ concentration was found. From the bottom of this layer to the ground, the $NO_y$ concentration remains approximately constant, and the particle number concentration shows some variations, but in general a block-shaped vertical profile is found. Aerosol size spectra for the intervals between and 900m–1800m, 1800m–2000m and 2000m–2400m indicate a bimodal log-normal size distribution with the modes of the distribution at 0.13µm and 2.0µm.

Seventy-two hour Hysplit (Stein et al., 2015) back-trajectories (not shown here) indicate that the air measured in these profiles had re-circulated over Mozambique, southern Zimbabwe and south-eastern Botswana before making its way in a south-easterly direction towards either Richards Bay or Nelspruit.

## 4   Radiative Transfer Modelling and Airmass Factor Calculation

It is clear that the optical properties of aerosols need to be included in the calculation of the air-mass factor (AMF, defined in Section 2) as highlighted by Leitão et al. (2010). In that study, several vertical profile scenarios are devised from chemistry-transport model output. For the present study, idealised scenarios representative of the Highveld are used to perform a sensitivity study using the SCIATRAN radiative transfer model (Rozanov et al., 2014). Our scenarios are based on two archetypal profile

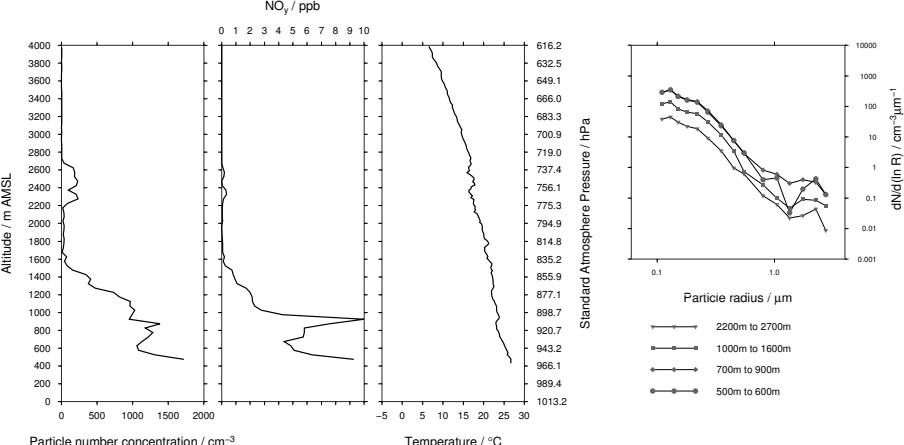

**Figure 2.** Vertical profiles of particle number concentration, in-situ $NO_y$ and temperature on 11 August 2007 between 11:01 and 11:40 UTC overhead the coastal town of Richards Bay. Average particle size spectra for altitude intervals of interest are plotted.

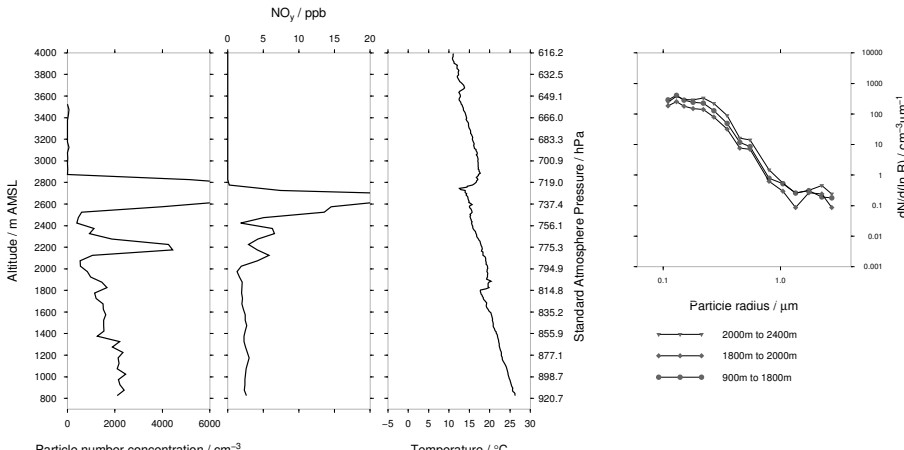

**Figure 3.** Vertical profiles of particle number concentration, in-situ $NO_y$ and temperature on 11 August 2007, overhead Nelspruit between 12:40 and 13:20 UTC. The aircraft landed at Nelspruit, so the profile is measured down to the surface. Average particle size spectra for altitude intervals of interest are plotted. The elevated layer between ca. 2900–2500m is due to a large forest fire plume.

shapes: a block-shaped profile where the concentration of aerosols and trace-gases is constant up to a certain height, and a profile where these concentrations decrease exponentially with height.

Measurements of aerosol optical thickness at 440nm (AOT), and retrievals of aerosol single-scattering albedo at 441nm (SSA or $\bar{\omega}$) over the Highveld are available in the AERONET Level 2.0 dataset from the sun photometer that was situated at the University of the Witwatersrand in Johannesburg during 2007, 2009, and 2011. Daily-mean statistics of these measurements are summarised in Table 1. The intention is not a detailed analysis of the AERONET record, but rather to determine reasonable AOT and $\bar{\omega}$ magnitudes for input into the radiative transfer model. Monthly-mean values of AOT for August are between

**Table 1.** A summary of AERONET Level 2.0 sun photometer measurements of daily-average aerosol optical thickness at 440nm at the University of the Witwatersrand in Johannesburg during late winter 2007, 2009, and 2011.

|  |  | AOT | | |
|  |  | 2007 | 2009 | 2011 |
| --- | --- | --- | --- | --- |
| | Min | 0.08 | 0.06 | 0.17 |
| July | Mean | 0.14 | 0.11 | 0.25 |
| | Max | 0.20 | 0.18 | 0.30 |
| | Min | 0.22 | 0.14 | 0.12 |
| Aug | Mean | 0.26 | 0.17 | 0.18 |
| | Max | 0.31 | 0.23 | 0.27 |
| | Min | 0.26 | 0.27 | 0.07 |
| Sept | Mean | 0.28 | 0.34 | 0.11 |
| | Max | 0.30 | 0.46 | 0.15 |

0.17 and 0.26 for the three years, however examining the data as a time-series (not shown here) we find that days with higher daily-mean AOT are associated with higher variability within that day, with values greater than 0.5 on some days. These high AOT values are likely associated with $SO_2$ plumes (Laakso et al., 2012) from industrial and household combustion processes which are also $NO_2$ sources. Biomass burning is also a source of both $NO_2$ and aerosols (Maenhaut et al., 1996; Eck, 2003), hence we choose representative aerosol optical thicknesses of 0.1, 0.3 and 0.5 and scale the vertical profile of scattering and absorption coefficients in our model runs appropriately. SSA retrievals are scarce in the Level 2.0 dataset, with no retrievals in 2007 during these months, and only three in 2009 ranging between 0.83 and 0.88. In 2011, the SSA values for the months of July to September ranged from 0.87 to 0.99.

Unlike the study of Leitão et al. (2010), wherein vertical profiles representative of urban and rural scenes with different VCD and AOT values were used, the present idealized scenarios are designed to all have the same $NO_2$ VCD of $20\,\mathrm{petamolec\,cm^{-2}}$ ($2\times10^{16}\,\mathrm{molec\,cm^{-2}}$), and as mentioned above AOT values of 0.1, 0.3 and 0.5 are used. The $NO_2$ volume mixing ratio (VMR) between the top of the block, and either the elevated layer above, or the top of the model grid is set at $1.0\times10^{-11}$ (10ppt), in order to avoid undefined block-AMFs in these parts of the vertical grid (a block-AMF is similar to an AMF, but for a subset of the total vertical column). Scenarios are introduced where an elevated layer of aerosols and trace gases are added to the profile shape, as has been observed during this and other field campaigns in the region. Since large portions of the Highveld are at altitude, the effect of a change in surface elevation from sea-level to 1400m above sea-level is evaluated. The twelve model scenarios' profile shapes are shown in Fig. 4.

The radiative transfer model (RTM) is run at a wavelength of 440nm, with the surface albedo set at 0.02, 0.05, 0.08 and 0.11. The solar zenith angle is varied from 45°–60° in steps of 5°. Aerosols are modelled using representative single-scattering albedos ($\bar{\omega}$) of 0.82, 0.90 and 0.98; and a Henyey-Greenstein phase function with an asymmetry parameter of 0.7

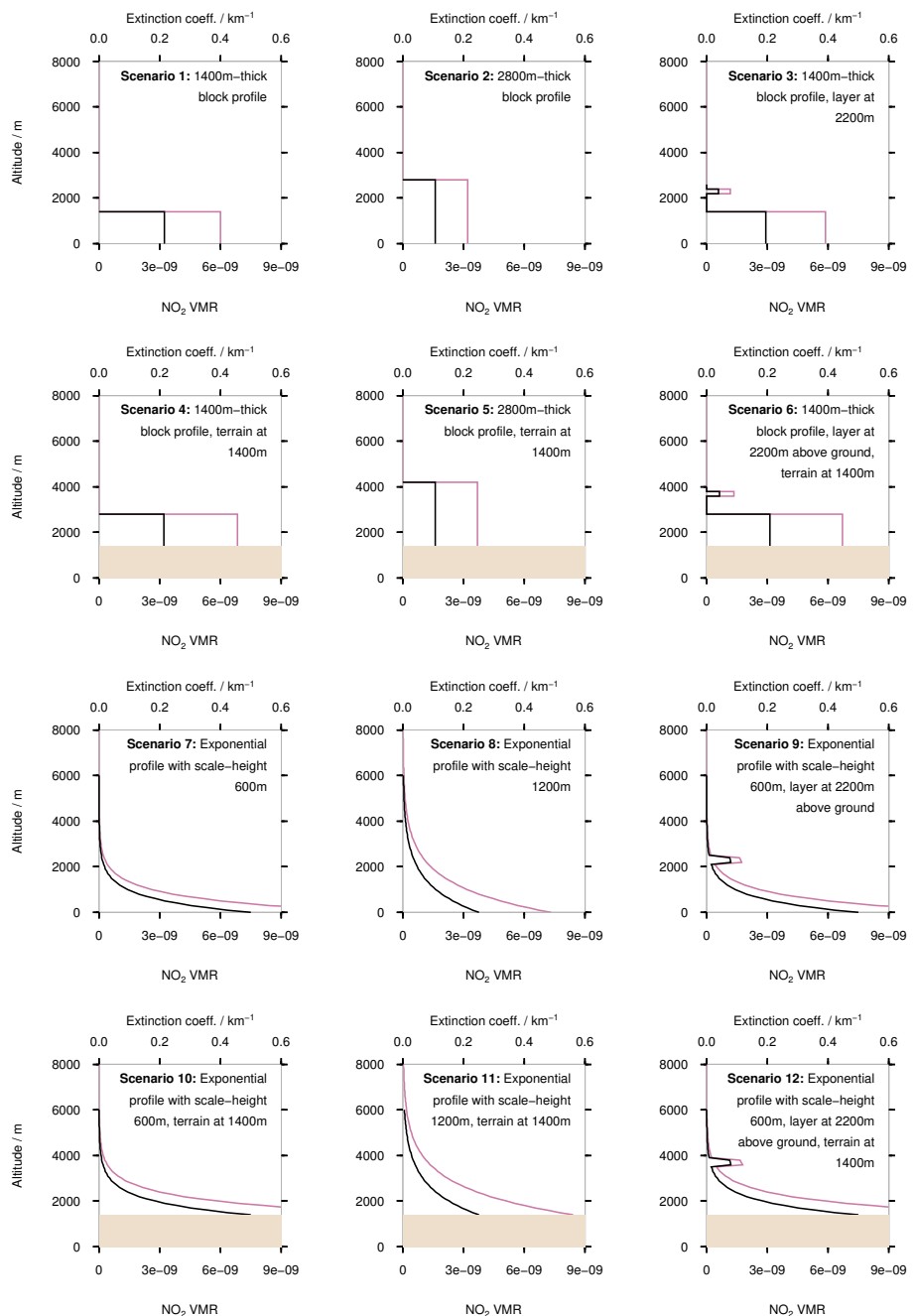

**Figure 4.** Profile shapes of aerosol extinction coefficient (black line) and $NO_2$ mixing ratio (purple line) for the twelve scenarios used in the radiative transfer model sensitivity study. The scenarios are designed to all have the same $NO_2$ VCD. Extinction and absorption coefficients are scaled to produce aerosol optical thicknesses of 0.1, 0.3, and 0.5. Terrain height is indicated by the shaded light-brown area. Note that the vertical grid used in the model extends up to 10000m

(Henyey and Greenstein, 1941). An altitude grid of 200m-thick layers from the surface up to 10000m is used. The aircraft altitude is fixed at 6000m above sea level in all scenarios. For the satellite viewing geometry, the observer is placed at the top of the altitude grid. Lambertian surface reflectance is assumed.

Calculated AMFs are summarised in Fig. 5 for scenarios with the surface elevation at sea-level, and in Fig. 6 for the scenarios with surface elevation at 1400m above sea-level. In both figures, results from model runs with the AOT set at 0.3 are plotted in orange; runs with AOT of 0.1 and 0.5 are in grey and blue-green respectively. Calculated AMFs at 0m surface elevation are summarised in Fig. 7: AMFs appear to be constrained between a minimum- and maximum-AMF surface. A similar plot for the results at 1400m would similarly constrain the AMFs between two surfaces. For a given combination of altitude, SZA and surface albedo, variation in the AMF is due to variation in the trace-gas and aerosol profile shapes, AOT, and aerosol single-scattering albedo. From the results shown in Figs. 5 and 6, it would appear that the influence of AOT on the range of likely AMFs for a given SZA and surface albedo is less than the combined influence of profile shape and SSA. Increased surface elevation broadens the uncertainty range. Although these scenarios are by no means exhaustive, they are representative of what may be found in the atmosphere above the Highveld, and allow the uncertainty in the AMF to be constrained.

## 5   Airborne DOAS measurements

The high spatial resolution (around 80m) nadir measurements from the iDOAS (Heue et al., 2008), when combined with a flight path following the satellite track, might be thought of as giving a transect of each satellite ground pixel. Two approaches are taken in order to make a comparison between measurements from the two platforms: a time-domain moving-average of the aircraft VCD measurement and a line-average of the high-resolution aircraft measurements within each satellite pixel.

Variability in the AMF for the aircraft measurements is constrained between the minimum- and maximum AMF surfaces described above in relation to Fig. 7. In order to determine the maximum- and minimum AMF, and hence the uncertainty in the VCD due to the profile shape and aerosol properties, successive linear interpolations between data points along SZA, surface albedo, and surface-elevation axes are performed for each iDOAS measurement. Solar zenith angle is calculated for the aircraft time and position using the pyEphem package (Rhodes, 2015). Surface albedos are sampled from the OMI albedo climatology (Kleipool et al., 2008), hence the spatial resolution of the albedo map is limited to half a degree. Surface elevation is sampled from the US Geological Survey GTOPO-30 global digital elevation model with spatial resolution of approximately 1km. For the flight maps presented below, the mean of the minimum- and maximum VCD is shown , with error-bars in the time-series plots defined by these VCD's, calculated using the maximum- and minimum AMF respectively. It should be noted that the mean does not represent the peak of the VCD probability density function as it might normally, and is used here as a shorthand to facilitate discussion. In reality there is only one correct vertical profile and associated AMF, so the true VCD is as likely to be the minimum or maximum as it is the mean.

In general cloud-free conditions were encountered during all of the flights. The exception to this is the flight on 11 August, where approximately one octa of thin cirrus cloud cover was observed above the aircraft, estimated to be at around 10000m. Based on the radiative-transfer-modelling study of Kokhanovsky and Rozanov (2009), the TVCD error caused by clouds in the

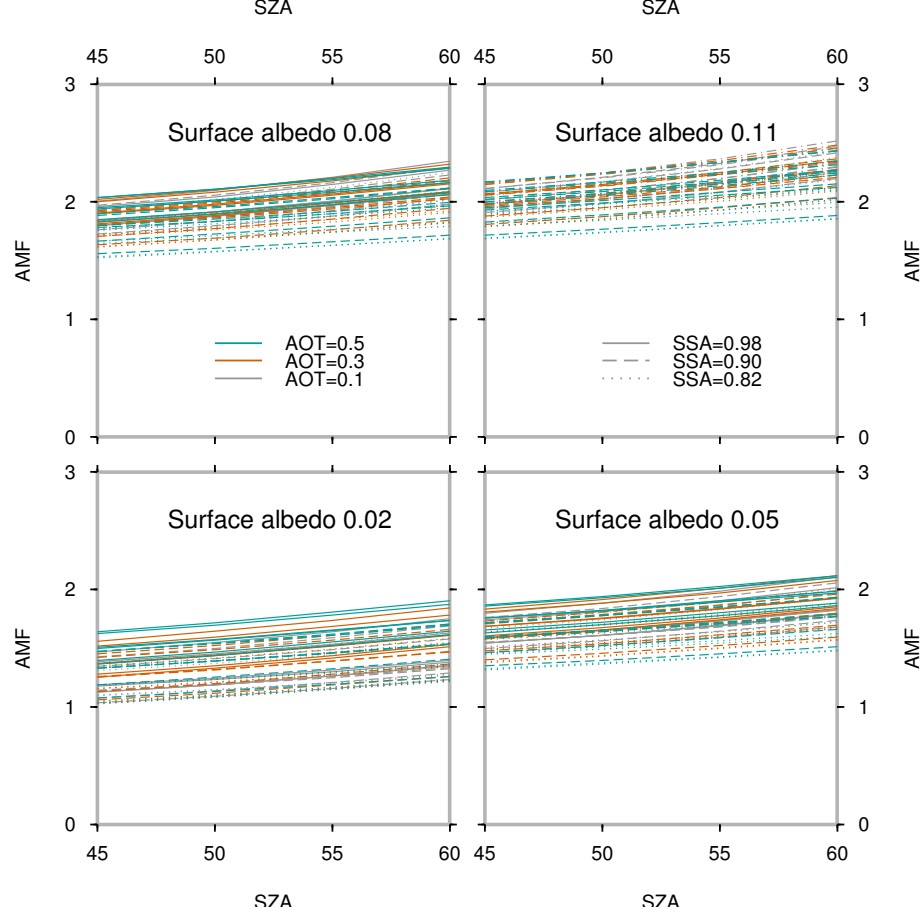

**Figure 5.** Calculated air-mass factors from the six low-surface-elevation (sea-level) scenarios outlined in Fig. 4, plotted as a function of solar zenith angle and surface albedo. A dotted line indicates $\bar{\omega}$=0.82, a dashed line $\bar{\omega}$=0.90 and a solid line $\bar{\omega}$=0.98. AOT is indicated by line colour: grey for AOT=0.1, orange for AOT=0.3, and blue-green for AOT=0.5.

OMI measurement is estimated to be less than -10%. The effect of such clouds on errors in the aircraft measurement is not quantified, however it is likely to be less than this.

**Flight on 9 August**

Figure 9 shows a map of tropospheric vertical column density (TVCD) from the Derivation of OMI tropospheric NO2
5 (DOMINO) V2.0 product for 9 August 2007, with the flight track and $NO_2$ VCD from the airborne instrument overlaid. The flight track is close to the eastern edge of the OMI pixel beneath it until approximately half-way between Johannesburg and Pretoria, where it crosses over to be on the western edge of the adjacent row of OMI pixels. Hourly-average wind directions from several weather stations are shown for the hour of the aircraft's overpass and the previous two hours. A time-series comparison of the airborne DOAS nadir $NO_2$ VCD with OMI TVCD on this day is shown in Fig. 10. In the time-series plot,

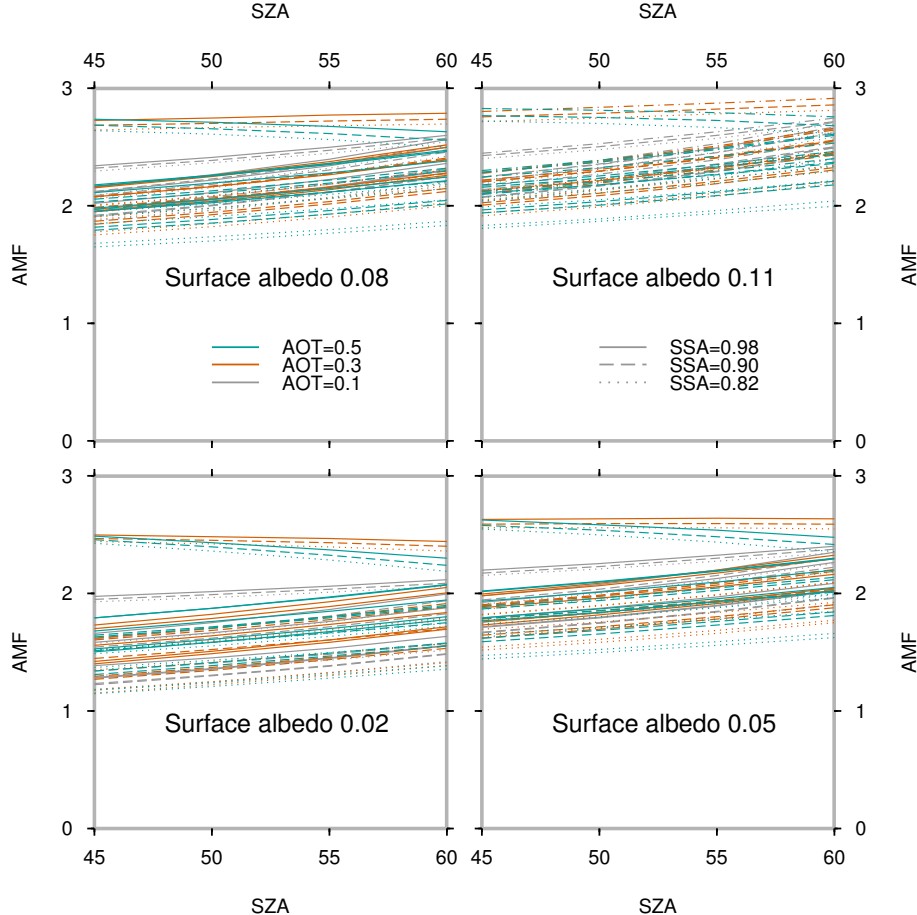

**Figure 6.** Calculated air-mass factors from the six high-surface-elevation (1400m) scenarios outlined in Fig. 4, plotted as a function of solar zenith angle and surface albedo. A dotted line indicates $\bar{\omega}$=0.82, a dashed line $\bar{\omega}$=0.90 and a solid line $\bar{\omega}$=0.98. AOT is indicated by line colour: grey for AOT=0.1, orange for AOT=0.3, and blue-green for AOT=0.5. At all surface albedos, the three upper curve families which stand out from the pack originate from Scenario 12.

airborne DOAS measurements are shown with error-bars representing the uncertainty in the AMF, along with the OMI pixel at aircraft nadir (orange) as well as one OMI-row to the west (blue) and east (yellow) of the aircraft. Line-averaged full-resolution iDOAS measurements within the OMI pixel at aircraft nadir are shown in grey, with one standard deviation in measured variability above and below the average shown by error-bars. The first subsidiary plot in Fig. 10 shows the surface elevation (grey) and surface albedo (orange) at aircraft nadir; the second subsidiary plot shows solar zenith angle at the aircraft's time and position (orange) as well as the minimum (grey) and maximum (cyan) AMF derived using the interpolation procedure described above. Aircraft time is indicated on the lower horizontal axis; the corresponding distance along the aircraft track is shown on the upper horizontal axis. Plots for subsequent flights present the measurements in a similar fashion.

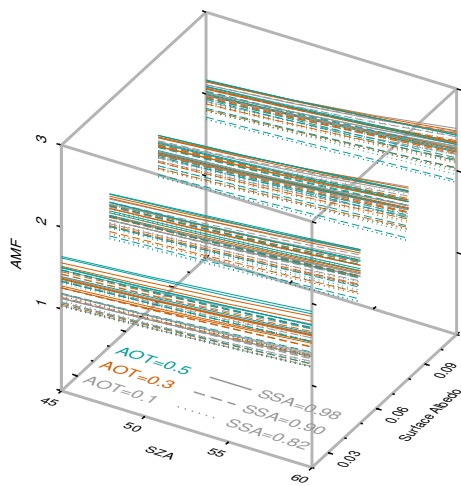

**Figure 7.** Calculated air-mass factors from the six low-surface-elevation scenarios outlined in Fig. 4, plotted as a function of solar zenith angle and surface albedo. A dotted line indicates $\bar{\omega}$=0.82, a dashed line $\bar{\omega}$=0.90 and a solid line $\bar{\omega}$=0.98. AOT is indicated by line colour: grey for AOT=0.1, orange for AOT=0.3, and blue-green for AOT=0.5. Variation in the AMF due to variations in aerosol properties and vertical profile shape are bounded by a minimum- and maximum-AMF surface at each surface elevation.

Figure 11 shows the airborne full-resolution iDOAS measurements for all the flights, averaged within the satellite pixel at aircraft nadir, compared with the $NO_2$ TVCD satellite product in that pixel. Error bars once again indicate one standard deviation in measured variability above and below the average. The regression lines are fitted through value pairs with satellite TVCD greater than $5\,\mathrm{petamolec\,cm^{-2}}$.

The airborne iDOAS measurements on 9 August included background areas, industrial plumes and urban areas. Several cases are identified from the time-series shown in Fig. 10 and the map in Fig. 9, and shown in Table 2. Weather-station data from Grootvlei indicates an hourly-average wind direction of 219° and $3.1\mathrm{m\,s^{-1}}$ for 12:00–13:00 UTC. The aircraft track is between 36km–48km downwind, and we might naïvely estimate that the plume is between approximately 3h–4h old when it was measured by the iDOAS.

Examining the cases presented in Table 2, with reference to Figs. 9 and 10, where the data are presented on a map and as a time-series respectively, we find background measurements by the iDOAS and OMI at 12:35 and 13:07. In both cases the peak, and line-averaged iDOAS VCD measurements are small. In the first case the OMI measurement is almost double the averaged iDOAS, in the second case it is half. The coefficient of variation (CV) of the background measurements is high, which indicates that the iDOAS is near its detection limit. At 12:45 and 12:55, horizontal gradients forming the shoulder of urban or industrial

plumes are measured. At 12:45 the averaged iDOAS value closely matches the OMI measurement, however this is by chance. The gradient across this OMI pixel is very steep, and the CV is high.

At 12:51 in the time-series the area between the two peaks is measured. In this case the OMI and average iDOAS measurements are within 20% of each other, and the CV is relatively low. The two main peaks in the time-series are at 12:47 and 12:53. The former appears to be the plume originating from the Vaal Triangle, the latter is measured near O.R. Tambo International

**Table 2.** Cases of background, urban, and plume measurements identified from the iDOAS measurements on 9 August 2007. Aircraft time is given in UTC. OMI nadir refers to the TVCD value in the OMI pixel at aircraft nadir. Similarly, OMI east and OMI west refer to the OMI pixel one row east and west of aircraft nadir respectively. iDOAS peak refers to the peak VCD within the OMI pixel at aircraft nadir. iDOAS average is the line-averaged iDOAS measurements within the OMI pixel at aircraft nadir, and iDOAS standard deviation is the iDOAS-measured variability within this OMI pixel expressed as a standard deviation. iDOAS CV is the coefficient of variation (or relative standard deviation) of the measured variability. All column densities are expressed in $\mathrm{petamolec\,cm^{-2}}$.

| | | OMI | | | iDOAS | | | |
| --- | --- | --- | --- | --- | --- | --- | --- | --- |
| Aircraft time | Description | nadir | west | east | peak | avg. | std. dev. | CV |
| 12:35 | Background | 2.0 | 3.4 | 3.5 | 4.0±1.1 | 0.97 | 0.94 | 0.9 |
| 12:45 | Plume | 11 | 34 | 7.8 | n/a | 10 | 4.4 | 0.4 |
| 12:47 | Plume | 15 | 24 | 14 | 36±12 | 27 | 3.6 | 0.1 |
| 12:51 | Peri-urban | 20 | 16 | 15 | n/a | 23 | 3.8 | 0.2 |
| 12:53 | Urban industrial | 25 | 30 | 14 | 99±23 | 56 | 21 | 0.4 |
| 12:55 | Urban industrial | 25 | 27 | 13 | n/a | 45 | 18 | 0.4 |
| 12:59 | Urban | 16 | 7.6 | 3.7 | 14±3.2 | 8.8 | 1.4 | 0.2 |
| 13:07 | Background | 0.8 | 1.3 | 2.0 | 4.9±0.9 | 1.6 | 1.0 | 0.6 |

Airport in Johannesburg. The Vaal Triangle peak measured by the iDOAS is $36\pm12\,\mathrm{petamolec\,cm^{-2}}$, more than double the value of 15 measured by OMI. The line-averaged iDOAS value is almost double the OMI measurement, and the peak serendipitously falls in the middle of the OMI pixel, so the CV is relatively low. Perhaps, since the aircraft is flying at the upwind edge of the nadir OMI pixel, the pixel to the west is a more appropriate comparison. In this case the iDOAS line-average is a better

match.

The peak value near the airport measured by the iDOAS at 12:53 is $99\pm23\,\mathrm{petamolec\,cm^{-2}}$. OMI fails to capture the magnitude of this peak. Horizontal $NO_2$ gradients do not always conveniently align themselves with the flight directions of satellites and aircraft, and therefore a similar gradient in the orthogonal direction is likely, the west- and east OMI pixels also do not capture the peak. The imaging swath of the iDOAS is quite narrow, around 1980m, from 4500m above ground, which

is too narrow to resolve the sort of gradient observed in the flight direction. A wider-swath imaging instrument may allow insights into the gradients within an OMI pixel. In this case, flying the aircraft perpendicular to the satellite track rather than along it might be a better flight strategy to optimise the use of the imaging swath, since this will place the airborne instrument swath along the short axis of the satellite pixel. The line-averaged OMI measurement of this peak is more than double the OMI value, with high variability within the pixel arising from the steep gradient.

In Fig. 11 the slope of the regression line for this flight is 2.2, indicating that for the industrial plumes and urban areas measured close to the sources during this flight, OMI substantially underestimates $NO_2$ VCD. This is likely to be because of poor horizontal dispersion of the plumes over the distance between the source and the measurement, however vertical dispersion may also play a role.

It is instructive to evaluate the potential air-mass factor error that might be made by assuming an incorrect vertical profile of $NO_2$. Several more radiative-transfer modelling scenarios are introduced, based on scenarios 11 and 12, i.e. with an exponentially-decreasing profile and surface elevation set at 1400m; some profiles with an elevated layer of $NO_2$ and some without. The scale height of the profiles is varied from 1400m to 200m, and radiative transfer calculations are done at a representative solar zenith angle of 55°. Once again air-mass factors for permutations of AOT of 0.1, 0.3, and 0.5, and SSA of 0.82, 0.90, and 0.98 are calculated. Results for aircraft- and satellite viewing geometry are presented in Fig. 8. The AMF increases for scenarios with an elevated $NO_2$ layer, the AMF increases as the vertical profile scale-height is decreased. In contrast, without such a layer, the AMF decreases as the scale-height is reduced. In the satellite viewing geometry, the behaviour is slightly different compared to the aircraft geometry, with a flattening off of the AMFs with scale-heights of 600m and 400m in the elevated-layer scenarios. This behaviour can likely be explained by examination of the block-AMFs for the two cases, however such analysis is beyond the scope of the present study.

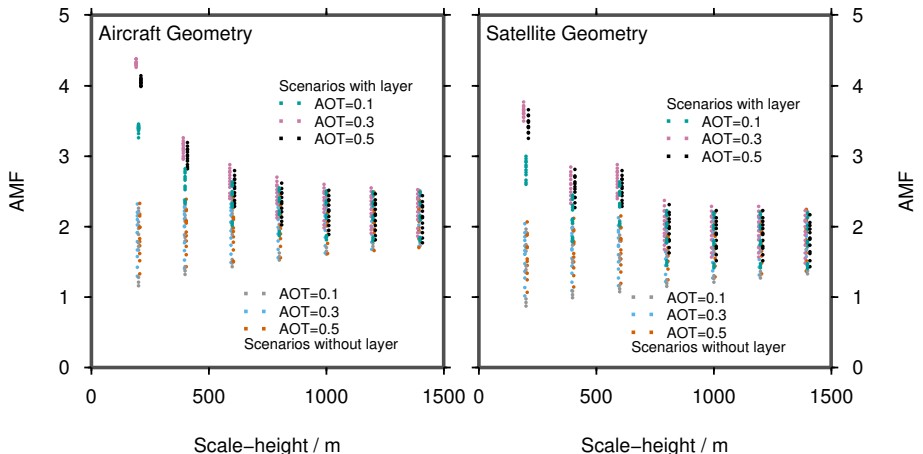

**Figure 8.** Air-mass factors at SZA=55° for scenarios based on an exponentially-decreasing vertical profile of $NO_2$ and aerosols, with scale heights varied from 1400m to 200m. These profiles are based on scenarios 11 and 12 from Fig. 4. Profiles with an elevated layer display an increasing trend of AMF with decreasing scale-height, those without the layer show a decrease in AMF.

We might estimate the VCD error arising from AMF uncertainty for the iDOAS using two profiles: the true profile at the spatial scale of the instrument, $P_{true}$ and the profile used in the AMF calculation $P_{prior}$, along with the associated AMFs: $AMF_{true}$ and $AMF_{prior}$. If $P_{prior}$ is an exponentially-decreasing profile with scale-height of 1000m either with- or without an elevated layer, according to Fig 8 $AMF_{prior}$ will lie between approximately 1.6 and 2.6. Close to a surface source of $NO_2$ $P_{true}$ might have a much smaller scale-height, for example 400m. In the case of a profile with an elevated layer, $AMF_{true}$ should be between 2.5 and 3.2. Using the mid-points of the uncertainty ranges of $AMF_{true}$ and $AMF_{prior}$, this will lead to a 26% overestimation of the VCD. In the case of $P_{true}$ having no elevated layer, $AMF_{true}$ will lie between approximately 1.2 and 2.3, leading to a 20% underestimation of the VCD from the use of $AMF_{prior}$.

In the case of a satellite measurement, a representative profile for the satellite pixel is likely to have a larger scale-height, since more background areas will be included in the measurement along with the surface source, and the discrepancy between $AMF_{prior}$ and $AMF_{true}$ will be less, but will behave in a similar manner to that described above. This highlights the importance of an improved $P_{prior}$ as the spatial resolution of the measurement improves. This has been implemented for OMI using a chemistry transport model at 0.667° by 0.5° resolution (compared to the global TM4 model's resolution of 3° by 2°) over eastern China, which resulted in an improved correlation with surface-based MAX-DOAS measurements (Lin et al., 2014). Future satellite missions which promise better spatial resolution (Veefkind et al., 2012) will require improved prior vertical profile estimates in order to avoid this divergence in the VCD uncertainty over surface sources.

**Table 3.** Cases of background and plume measurements identified from the iDOAS measurements on 11 August 2007. Aircraft time is given in UTC. OMI nadir refers to the TVCD value in the OMI pixel at aircraft nadir. Similarly, OMI east and OMI west refer to the OMI pixel one row east and west of aircraft nadir respectively. iDOAS peak refers to the peak VCD within the OMI pixel at aircraft nadir. iDOAS average is the line-averaged iDOAS measurements within the OMI pixel at aircraft nadir, and iDOAS standard deviation is the iDOAS-measured variability within this OMI pixel expressed as a standard deviation. iDOAS CV is the coefficient of variation (or relative standard deviation) of the measured variability. All column densities are expressed in $\mathrm{petamolec\,cm^{-2}}$.

| | | OMI | | | iDOAS | | | |
| Aircraft time | Description | nadir | west | east | peak | avg. | std. dev. | CV |
| --- | --- | --- | --- | --- | --- | --- | --- | --- |
| 11:52 | Background | 2.4 | 3.1 | 4.0 | n/a | 2.5 | 1.1 | 0.44 |
| 12:14 | Dispersed Plume | 12 | 11 | 18 | 14±3.0 | 8.7 | 1.3 | 0.15 |
| 12:22 | Dispersed Plume | 31 | 21 | 36 | 41±8.1 | 32 | 2.2 | 0.06 |
| 12:25 | Dispersed Plume | 37 | 30 | 39 | 44±9.4 | 36 | 3.2 | 0.09 |
| 12:32 | Dispersed Plume | 28 | 36 | 17 | 48±11 | 40 | 3.1 | 0.08 |

**Flight on 11 August**

On 11 August, the aircraft measured background values, as well as several cases of plumes originating from industrial facilities on the Highveld. Wind measurements at Camden of $8.6\mathrm{m\,s^{-1}}$ at 12:00 UTC allow a naïve estimate of the plume age from approximately 5h for the closest power station (approximately 145km as the wind blows) to 12h for the Vaal Triangle and the city of Johannesburg, approximately 360km away. In reality, the plumes are probably older than this, since the wind speeds in the 12h prior to the flight were lower, and this is confirmed by looking at a back-trajectory (not shown here) for air parcels over Swaziland for this day. This is slightly further downwind than the Vaal Triangle plume measured on 9 August, and clearly much further downwind than the $NO_2$ plume measured overhead the city of Johannesburg on that day, which is measured at the source.

These cases are enumerated in Table 3, which should be read in conjunction with Figs. 12 and 13. At 11:52, the iDOAS measured background $NO_2$ between Richards Bay and Swaziland. The line-averaged iDOAS matches the satellite measurement.

The measured variability is low, although the coefficient of variation (CV) is higher than for the other cases in Table 3. As is the case on 9 August, this again indicates that the iDOAS is operating close to its detection limit. The plume from the Vaal Triangle is measured at 12:14, in this case the OMI measurement is higher than the iDOAS line-average value. The variability is similar to the background measurement, giving a much lower CV. At 12:22 and 12:25 the southern locus of a mega-plume appearing to originate from the Highveld is measured. In these cases, the line-averaged iDOAS agrees well with the satellite measurement and the CV within each OMI pixel is very low. At 12:32 what appears to be the northern locus of the plume is measured by the iDOAS, with peak $NO_2$ VCD of $48\pm11$ petamolec cm$^{-2}$. This peak appears in the OMI measurement at a considerably lower magnitude, although the upwind (west) pixel is more comparable with the iDOAS peak and average values. The coefficient of variation in this case is very low.

The reason for the dual locus of this plume is not clear; it is perhaps related to the topography where this plume appears to be on either side of a ridge. These two plume loci are resolved by the OMI satellite at aircraft nadir and one line upwind; one line downwind the two loci appear to have merged into one. The structure seen in the satellite image in Fig. 12 of a southern plume advecting from the Vaal Triangle, and a northern plume from Johannesburg and the cluster of power stations on the eastern Highveld is seen frequently while browsing through the OMI record. In Fig. 11 the regression line for this flight has a slope of 1.1, indicating that OMI is better able to capture the shallower $NO_2$ VCD gradients in a dispersed plume better than in the narrow plumes near to emissions sources, and urban areas measured on 9 August.

**Table 4.** Cases of background and plume measurements identified from the iDOAS measurements on 18 August 2007. Aircraft time is given in UTC. OMI nadir refers to the TVCD value in the OMI pixel at aircraft nadir. Similarly, OMI east and OMI west refer to the OMI pixel one row east and west of aircraft nadir respectively. iDOAS peak refers to the peak VCD within the OMI pixel at aircraft nadir. iDOAS average is the line-averaged iDOAS measurements within the OMI pixel at aircraft nadir, and iDOAS standard deviation is the iDOAS-measured variability within this OMI pixel expressed as a standard deviation. iDOAS CV is the coefficient of variation (or relative standard deviation) of the measured variability. All column densities are expressed in petamolec cm$^{-2}$.

| | | OMI | | | iDOAS | | | |
|---|---|---|---|---|---|---|---|---|
| Aircraft time | Description | nadir | west | east | peak | avg. | std. dev. | CV |
| 12:41 | Dispersed plume | 8.2 | 1.8 | 10 | $22\pm5.2$ | 23 | 1.9 | 0.08 |
| 12:45 | Plume | 29 | 32 | 23 | $61\pm15$ | 43 | 6.9 | 0.16 |
| 12:54 | Plume | 39 | 35 | 32 | $67\pm16$ | 53 | 5.2 | 0.10 |
| 13:01 | Plume | 40 | 30 | 24 | $84\pm21$ | 63 | 11 | 0.17 |
| 13:03 | Plume | 44 | 43 | 34 | $79\pm20$ | 53 | 7.2 | 0.14 |

**Flight on 18 August**

The flight on 18 August, shown in Fig. 14, routed close to the power stations on the Eastern Highveld. Several plumes are identified from the iDOAS measurement time-series in Fig. 15 and the map in Fig. 14 and summarised in Table 4. The aircraft

track was approximately 55km downwind of Majuba, 110km downwind of Tutuka power station and 140km downwind of Secunda. The Vaal Triangle was approximately 250km upwind of the aircraft. Weather station measurements from Tutuka indicate windspeeds of $2.1\,\mathrm{m\,s^{-1}}$, giving a naïve estimate of plume age ranging from 7h from Majuba to 18h from Secunda. All of the cases in Table 4 are plume measurements. In all the cases both the peak and the line-averaged iDOAS measurements are higher than the OMI measurement. The coefficients of variation are generally intermediate between those found on 9 and 11 August. The regression through OMI-pixel-averaged aircraft measurements compared with the OMI product from 18 August in Fig. 11 has a slope of 1.3. This reflects the lower $NO_2$ VCDs measured by OMI in comparison to the iDOAS, described above in relation to the time-series in Fig. 15.

Comparing the three flights that were performed to track the Aura satellite, we find a relation between the slope of the regression line and the distance of the measurement from the source. For the flight on 9 August, which passed approximately 40km downwind of the Vaal Triangle and directly overhead Johannesburg the slope of the regression line is 2.2. Measurements on 11 August, between $145-360$km downwind of sources give a slope of 1.1 and the measurements on 18 August, where the aircraft was $55-150$km downwind of major sources gives an intermediate slope of 1.3. There appears to be a similar relation between distance downwind and coefficient of variation within each OMI pixel. This is what we would expect, since a more dispersed plume would have lower horizontal gradients, and hence a better match between the satellite and iDOAS. Plume dispersion occurs by turbulent mixing in the boundary layer, which is related to surface topography and instability (i.e. thermals). Both of these are spatial features of the landscape hence the relationship of degree of mixing with distance downwind, rather than the time taken to travel the distance. An additional effect evident from the modelling results in Fig. 8 is that AMF errors will be less when there is more vertical dispersion, and therefore a profile with a higher scale-height.

Comparison studies of ground-based multi-axis DOAS (MAX-DOAS) instruments with satellite measurements have given mixed results. Some studies (Irie et al., 2008; Hains et al., 2010) show MAX-DOAS results consistently lower than OMI. Kanaya et al. (2014) shows DOMINOv2 biases of up to 50% lower than the MAX-DOAS, although the bias improves when only remote surface sites are considered. This is attributed to both horizontal inhomogeneity within the OMI pixels and the inability of OMI to observe $NO_2$ close to the surface.

It is clear from inspection of the peak iDOAS measurements in Tables 2, 3, and 4 that our approach of using a range of AMFs to calculate the VCD results in a variability in the VCD that scales with SCD. This is simply a mathematical effect that is obvious from Eq. 2. It is interesting to note in the OMI measurements that the downwind pixels (yellow in the time-series plots) reproduce the plume structures observed in the aircraft-nadir OMI pixels (bold orange), with in general lower VCD. This would indicate a steady decrease in the amount of $NO_2$ in the air, probably due to chemical conversion into another species.

**Flight on 14 August**

A comparison between the airborne iDOAS and SCIAMACHY can be made from the flight on 14 August. Several cases identified in Fig. 16 and 17 are summarised in Table 5. The background measurements at 08:17 and 08:27 by SCIAMACHY and iDOAS are similar, and once again the coefficient of variation (CV) is high. Plumes are measured at 07:56, 08:01, and 08:08. In each case, the average iDOAS is lower than the SCIAMACHY measurement, although the CV is higher than for

other flights. At 08:12 the iDOAS measured the gradient on the shoulder of a plume, in this case the average iDOAS and the SCIAMACHY measurement are close, and the CV is similar to the gradient measurement at 12:55 on 9 August (shown in Table 2)

Examining Fig. 11 we see that the slope of the regression line between SCIAMACHY and the averaged iDOAS is less
than unity, indicating that the iDOAS underestimates relative to SCIAMACHY. Comparing the performance of OMI and SCIAMACHY against the iDOAS, one would expect OMI, given its higher spatial resolution, to be better able to capture the peak VCD's in the narrow plumes found on the Highveld. Given that we have only one SCIAMACHY comparison, it is difficult to draw any firm conclusion here.

**Table 5.** Cases of background and plume measurements identified from the iDOAS measurements on 14 August 2007. Aircraft time is given in UTC. SCIA nadir refers to the TVCD value in the SCIAMACHY pixel at aircraft nadir. iDOAS peak refers to the peak VCD within the OMI pixel at aircraft nadir. iDOAS average is the line-averaged iDOAS measurements within the SCIAMACHY pixel at aircraft nadir, and iDOAS standard deviation is the iDOAS-measured variability within this SCIAMACHY pixel expressed as a standard deviation. iDOAS CV is the coefficient of variation (or relative standard deviation) of the measured variability. All column densities are expressed in $petamolec\,cm^{-2}$.

| Aircraft time | Description | SCIA nadir | iDOAS peak | iDOAS avg | iDOAS std deviation | iDOAS CV |
|---|---|---|---|---|---|---|
| 07:56 | Dispersed plume | 19 | $21\pm2.7$ | 12 | 4.6 | 0.38 |
| 08:01 | Plume | 19 | $34\pm8.9$ | 14 | 8.4 | 0.60 |
| 08:08 | Plume | 10 | $23\pm5.6$ | 9.0 | 5.5 | 0.61 |
| 08:12 | Plume / background | 7.6 | $12\pm2.9$ | 4.8 | 2.2 | 0.46 |
| 08:17 | Background | 6.3 | $7.8\pm1.9$ | 2.8 | 1.5 | 0.53 |
| 08:27 | Background | 4.1 | $8.7\pm2.0$ | 3.4 | 1.5 | 0.44 |

## 6 Conclusions

Four research flights were performed over the Highveld region of South Africa during August 2007 using an airborne imaging DOAS instrument to measure $NO_2$ column densities, a Particle Measurement Systems PCASP probe to measure aerosol size distribution and number density, and a chemiluminescence in-situ $NO_y$ instrument. These flights were planned to co-incide with overpasses of the OMI and SCIAMACHY satellite instruments, with the aircraft measurements within an hour of the satellite. Each flight included a vertical profile measurement at the beginning and end of the airborne DOAS measurement
segment.

Vertical profile measurements of $NO_y$ and aerosol particle number concentrations, although compromised by problems with instruments and limitations due to flight safety requirements, reveal several features consistent with profiles reported in the literature. Profile shapes can be approximated by a block-shape and an exponentially-decreasing profile of trace-gas and aerosol concentration, and elevated layers of enhanced concentration are sometimes present over the Highveld. Observations

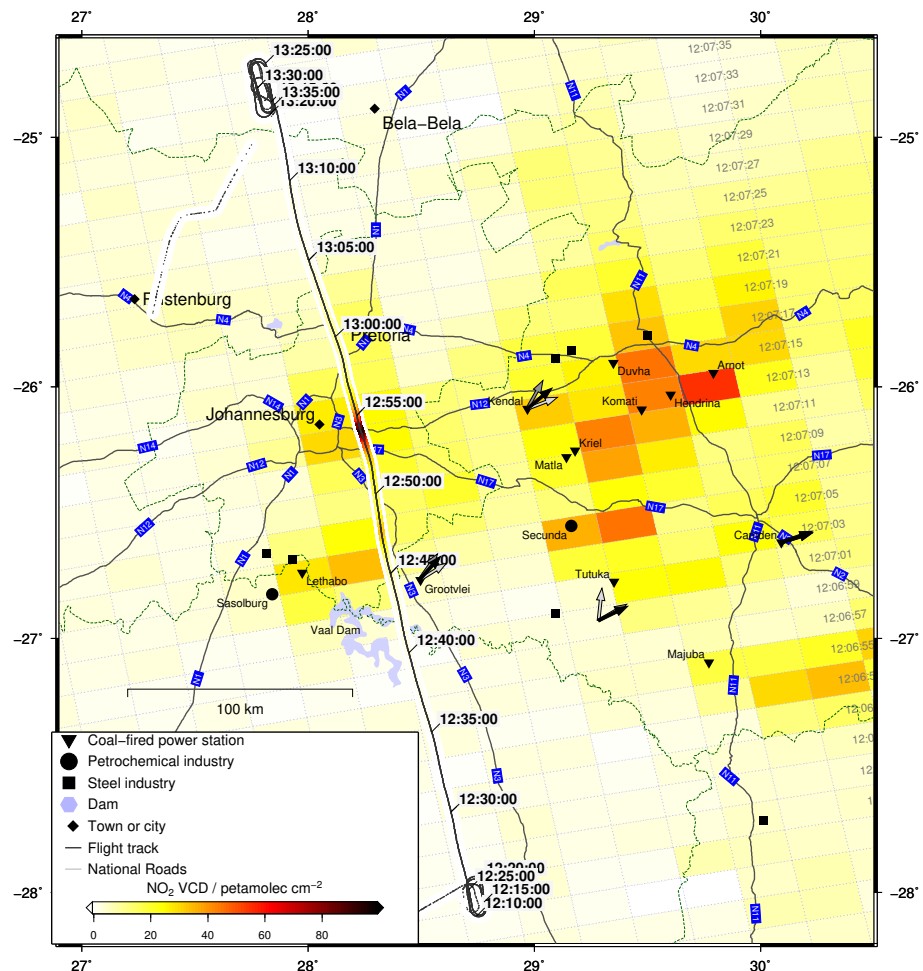

**Figure 9.** A comparison of the 10-second moving average airborne DOAS $NO_2$ vertical column densities (indicated by colour along the flight track) with OMI DOMINO V2.0 (coloured rectangles) on 9 August 2007. UTC aircraft time is indicated every five minutes along the flight track (black line running from approximately 28°S, 28.8°E to 24.8°S, 27.8°E). UTC satellite time is shown for each row. Hourly-average wind directions for several weather stations are shown for the hours up to 13:00 UTC (black arrow), 12:00 UTC (dark grey arrow) and 11:00 UTC (light grey arrow).

of aerosol optical thickness and single-scattering albedo from AERONET during July to September 2007 and 2009 are used to determine representative values for these parameters.

These observations are used to devise a number of scenarios, which are used in a sensitivity study using the SCIATRAN radiative transfer model. A minimum- and maximum air-mass factor is found for a given combination of surface elevation and albedo, and solar zenith angle. The difference between the minimum and maximum air-mass factor represents uncertainty due to the profile shape and aerosol properties. These air-mass factor estimates are used to calculate vertical column densities

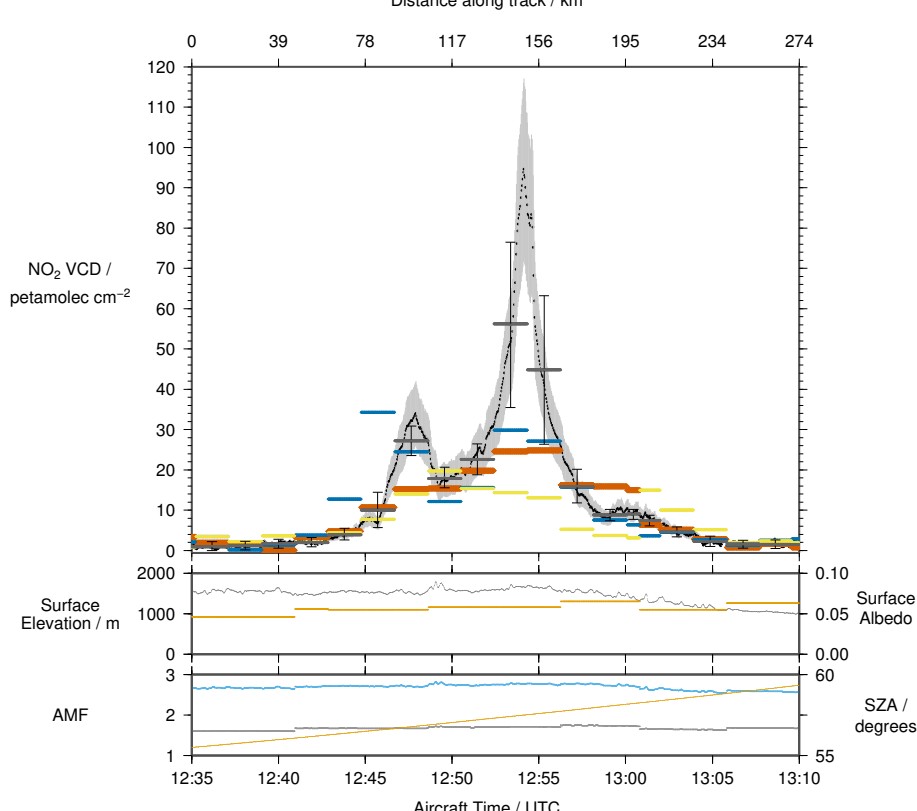

**Figure 10.** A timeseries of the airborne DOAS $NO_2$ vertical column densities on 9 August 2007, DOMINO V2.0 at aircraft nadir (orange) and one OMI line west (blue) and east (yellow) of aircraft nadir. Aircraft measurements averaged over the length of each OMI pixel are shown in grey, with one standard deviation in measured variability above and below the average indicated with error-bars. Surface elevation (grey) and surface albedo (orange) are shown in the first sub-plot. The second sub-plot shows the minimum and maximum AMF estimates (grey and blue), and the solar zenith angle (orange) at aircraft time and position.

from the slant-column densities measured by the iDOAS instrument . These are then compared to satellite tropospheric $NO_2$ products from OMI and SCIAMACHY.

The present approach to quantification of the uncertainty in the air-mass factor, and hence vertical column density, implies that the uncertainty in the vertical column density scales with the magnitude of the slant column density, in cases where the
5   uncertainty in the air-mass factor is the dominant source of error.

Analysis of air-mass factors from vertical profiles with a variety of scale-heights indicates that uncertainty increases as the scale-height decreases, as may be the case when making high spatial resolution measurements close to a surface source of $NO_2$. Indeed, the presence or absence of an elevated layer, leads to uncertainty in the sign of the error in the air-mass factor. This implies that air-mass factor errors will be less further downwind of sources, where vertical mixing within the boundary
10   layer has taken place.

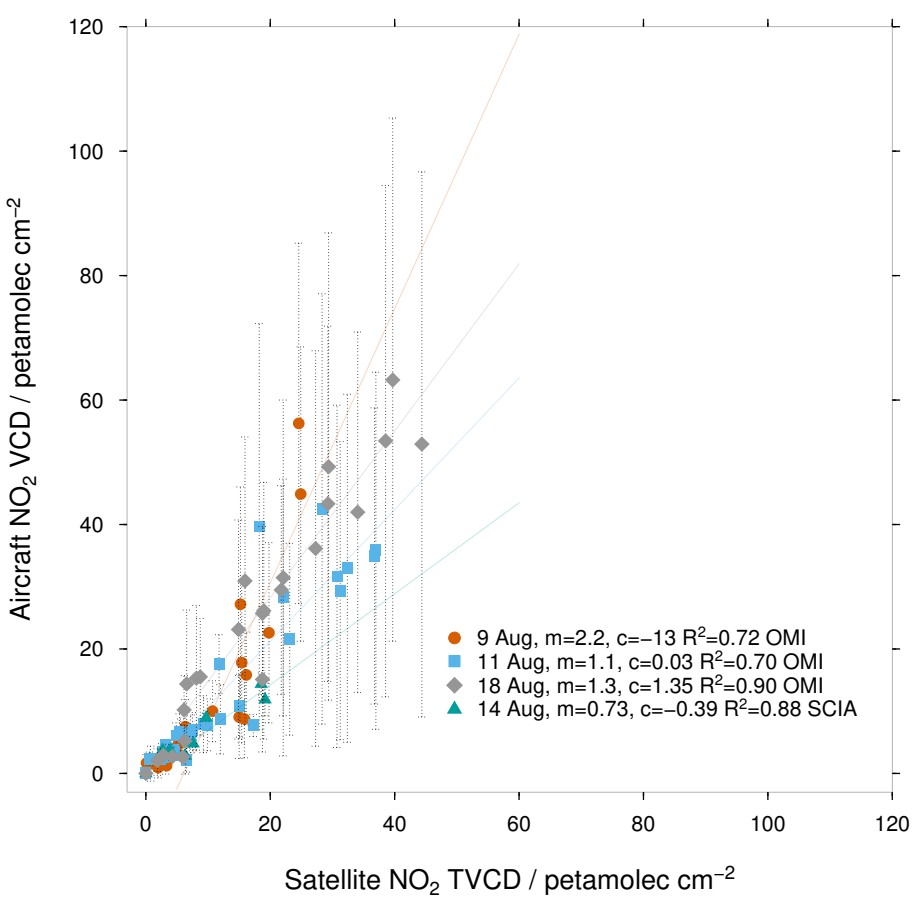

**Figure 11.** OMI and SCIAMACHY measurements compared with co-located line-averaged aircraft measurements. Aircraft iDOAS VCD is calculated using the mean of the minimum- and maximum-AMF calculated for the high-resolution measurement. Error-bars indicate the minimum- and maximum iDOAS VCD found within the satellite pixel. In the inset figure, a linear regression line is fitted through all background measurements less than $5\,\mathrm{petamolec\,cm^{-2}}$. In the main figure, the regression lines are fitted through measurements greater than $5\,\mathrm{petamolec\,cm^{-2}}$ for each flight. The slope (m) of the regression line for each day is indicated, as well as the y-intercept (c) and the regression coefficient ($R^2$).

The airborne DOAS instrument's much higher spatial resolution, even when averaged using a moving average on a spatial scale of approximately 1.2km, reveals spatial gradients in $NO_2$ that are much steeper than those observed by the satellites. Large-scale features are resolved by the satellites, however peak $NO_2$ vertical column densities observed by the aircraft close to urban and industrial sources are in some cases more than twice the satellite measurement. The performance of the satellite measurement was found to better for more dispersed plumes, measured further downwind from the source.

For measurements further than approximately 150km downwind, the agreement between the aircraft and OMI is within the margin of error of approximately 40% arising from uncertainty in the air-mass factor. This is due to the decrease in horizontal $NO_2$ gradients from turbulence in the mixed layer, which is dependent on spatial features such as surface topography and the

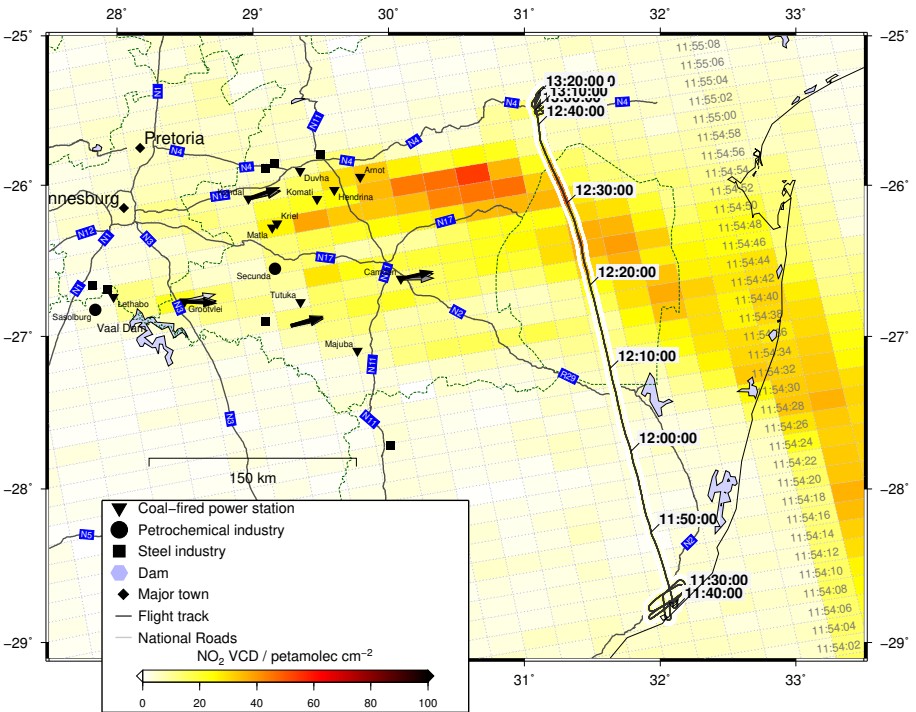

**Figure 12.** A comparison of the airborne DOAS 10-second moving average $NO_2$ vertical column densities between Richards Bay and Nelspruit with OMI DOMINO V2.0 on 11 August 2007. Data presentation is as in Fig. 9.

characteristics of thermals during the day. As such the agreement between the spatially-averaged iDOAS $NO_2$ VCD and the satellite product improves with distance, better than it does with time, downwind of the source.

Inspection of OMI Level-2 satellite images allow plumes from certain point sources on the Highveld to be identified. In other cases, plumes from areas containing several point sources, or effective area sources such as the city of Johannesburg can be identified. During the winter, these plumes are sufficiently stable that they retain their structure for several hundred kilometers downwind. This leads to a northern- and southern plume being visible on the satellite image, corresponding to sources on the eastern Highveld and the Vaal Triangle.

The high spatial resolution of the airborne instrument reveals spatial features in the $NO_2$ distribution that are not visible even at the relatively high resolution of the OMI sensor. Upcoming satellite missions such as TROPOMI (Veefkind et al., 2012), which have a higher spatial resolution than OMI promise to reveal small-scale features using daily measurements from space, however improved prior vertical profiles will be needed to constrain the air-mass factor uncertainty close to surface sources.

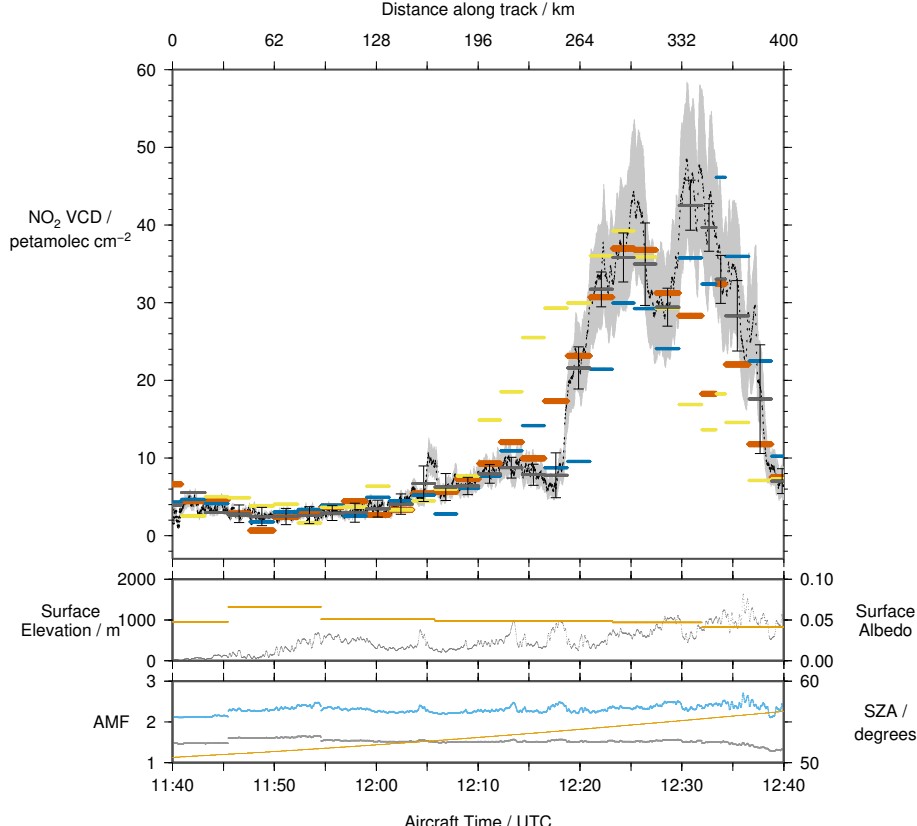

**Figure 13.** A timeseries of the airborne DOAS $NO_2$ vertical column densities on 11 August 2007, DOMINO V2.0 at aircraft nadir (orange) and one OMI line west (blue) and east (yellow). Aircraft measurements averaged over the length of each OMI pixel are shown in grey. Sub-plots are as described for Fig. 10.

*Acknowledgements.* GTOPO-30 DEM data is available from the U.S. Geological Survey. Funding was received from Eskom SOC Ltd. for this project. Thanks to the South African Weather Service, and aircraft crews for support during field campaigns. We acknowledge the free use of tropospheric $NO_2$ column data from the OMI sensor from www.temis.nl. Alexander Kokhanovsky acknowledges support of the excellence centre from applied mathematics and theoretical physics within MEPhI Academic Excellence Project (contract No. 02.a03.21.0005,27.08.2013)

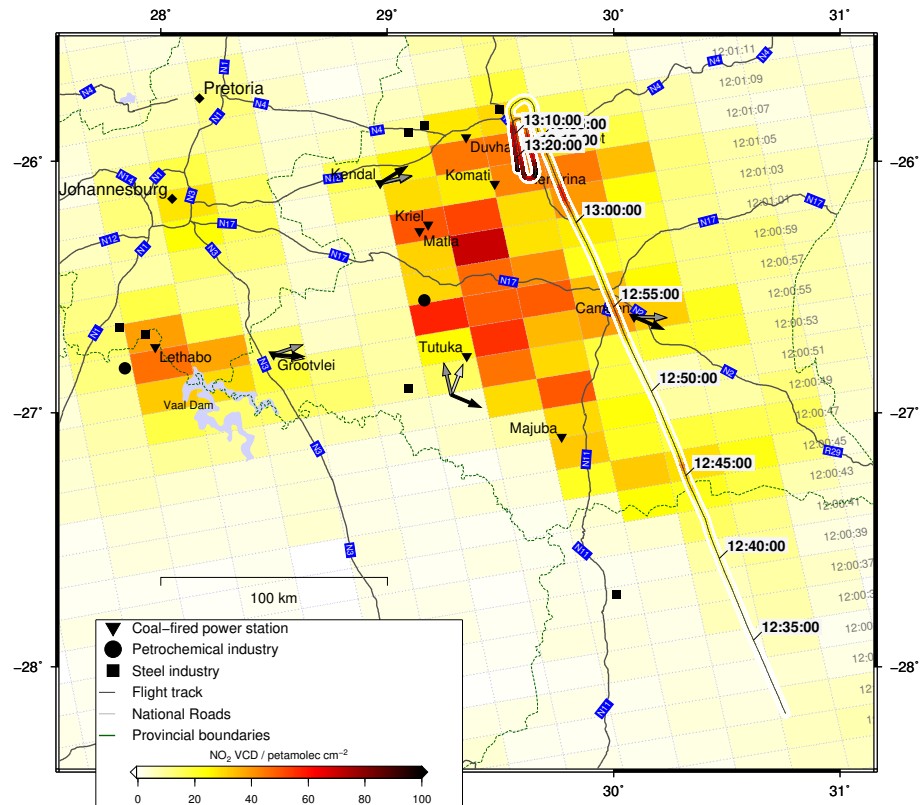

**Figure 14.** A comparison of the airborne DOAS 10-second moving average NO$_2$ vertical column densities with DOMINO V2.0 on 18 August 2007. Data presentation is as in Fig. 9

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

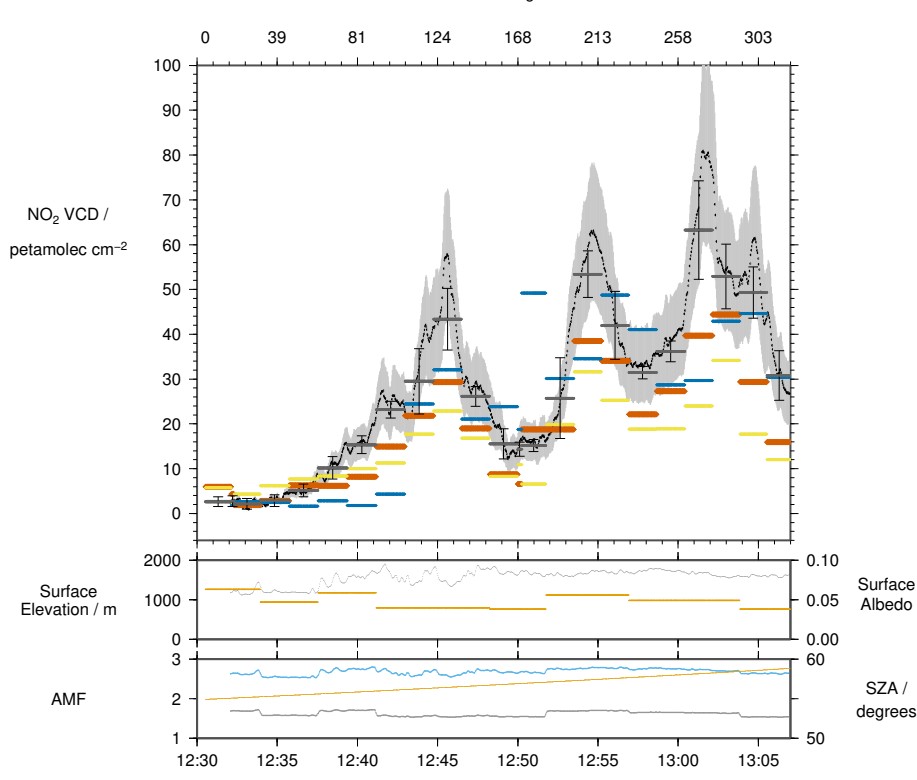

**Figure 15.** A timeseries of the airborne DOAS NO$_2$ vertical column densities on 18 August 2007, DOMINO V2.0 at aircraft nadir (orange) and one OMI line west (blue) and east (yellow). Aircraft measurements averaged over the length of each OMI pixel are shown in grey. Sub-plots are as described for Fig. 10

Radiative Transfer, 61, 509–517, doi:10.1016/S0022-4073(98)00037-5, http://linkinghub.elsevier.com/retrieve/pii/S0022407398000375, 1999.

Collett, K. S., Piketh, S. J., and Ross, K. E.: An assessment of the atmospheric nitrogen budget on the South African Highveld, South African Journal of Science, 106, 1–9, doi:10.4102/sajs.v106i5/6.220, http://www.sajs.co.za/index.php/SAJS/article/view/220, 2010.

5 Dentener, F. J., van Weele, M., Krol, M., Houweling, S., and van Velthoven, P.: Trends and inter-annual variability of methane emissions derived from 1979-1993 global CTM simulations, Atmospheric Chemistry and Physics, 3, 73–88, doi:10.5194/acpd-2-249-2002, 2003.

Eck, T. F.: Variability of biomass burning aerosol optical characteristics in southern Africa during the SAFARI 2000 dry season campaign and a comparison of single scattering albedo estimates from radiometric measurements, Journal of Geophysical Research, 108, doi:10.1029/2002JD002321, http://www.agu.org/pubs/crossref/2003/2002JD002321.shtml, 2003.

10 Gottwald, M., Bovensmann, H., and (Eds): SCIAMACHY, Exploring the Changing Earth's Atmosphere, Springer Dordrecht Heidelberg London New York, doi:10.1007/978-90-481-9896-2, 2006.

Greenblatt, G., Orlando, J., Burkholder, J., and Ravishankara, A. R.: Absorption measurements of Oxygen between 330nm and 1140nm, Journal of Geophysical Research, 95, 18 577–18 582, 1990.

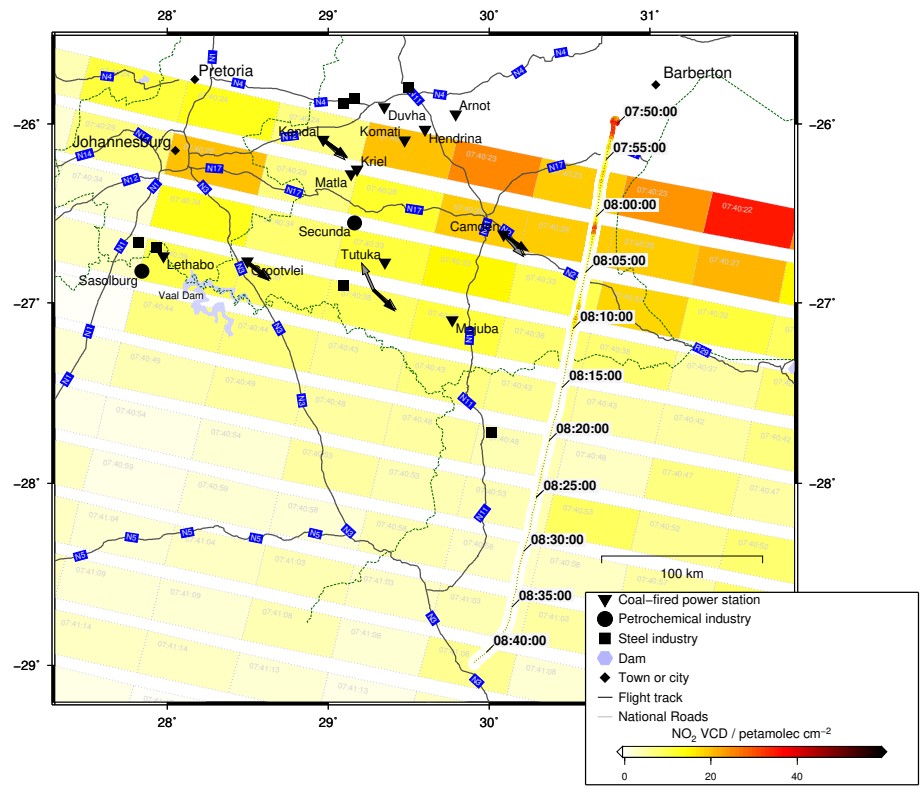

**Figure 16.** A comparison of the airborne DOAS 10-second moving average $NO_2$ vertical column densities with SCIAMACHY on 14 August 2007. Data presentation is as in Fig. 9, however since SCIAMACHY is a whiskbroom instrument, the time for each satellite pixel is shown in each pixel. SCIAMACHY overpass was at around 07:40 UTC.

Hains, J. C., Boersma, K. F., Kroon, M., Dirksen, R. J., Cohen, R. C., Perring, A. E., Bucsela, E., Volten, H., Swart, D. P. J., Richter, A., Wittrock, F., Schoenhardt, A., Wagner, T., Ibrahim, O. W., van Roozendael, M., Pinardi, G., Gleason, J. F., Veefkind, J. P., and Levelt, P. F.: Testing and improving OMI DOMINO tropospheric NO 2 using observations from the DANDELIONS and INTEX-B validation campaigns, Journal of Geophysical Research, 115, 1–20, doi:10.1029/2009JD012399, http://www.agu.org/pubs/crossref/2010/2009JD012399.shtml, 2010.

Haywood, J. M.: Comparison of aerosol size distributions, radiative properties, and optical depths determined by aircraft observations and Sun photometers during SAFARI 2000, Journal of Geophysical Research, 108, doi:10.1029/2002JD002250, http://www.agu.org/pubs/crossref/2003/2002JD002250.shtml, 2003a.

Haywood, J. M.: The mean physical and optical properties of regional haze dominated by biomass burning aerosol measured from the C-130 aircraft during SAFARI 2000, Journal of Geophysical Research, 108, doi:10.1029/2002JD002226, http://www.agu.org/pubs/crossref/2003/2002JD002226.shtml, 2003b.

Henyey, L. and Greenstein, J.: Diffuse radiation in the galaxy, Astrophysical Journal, 93, 70–83, doi:10.1086/144246, 1941.

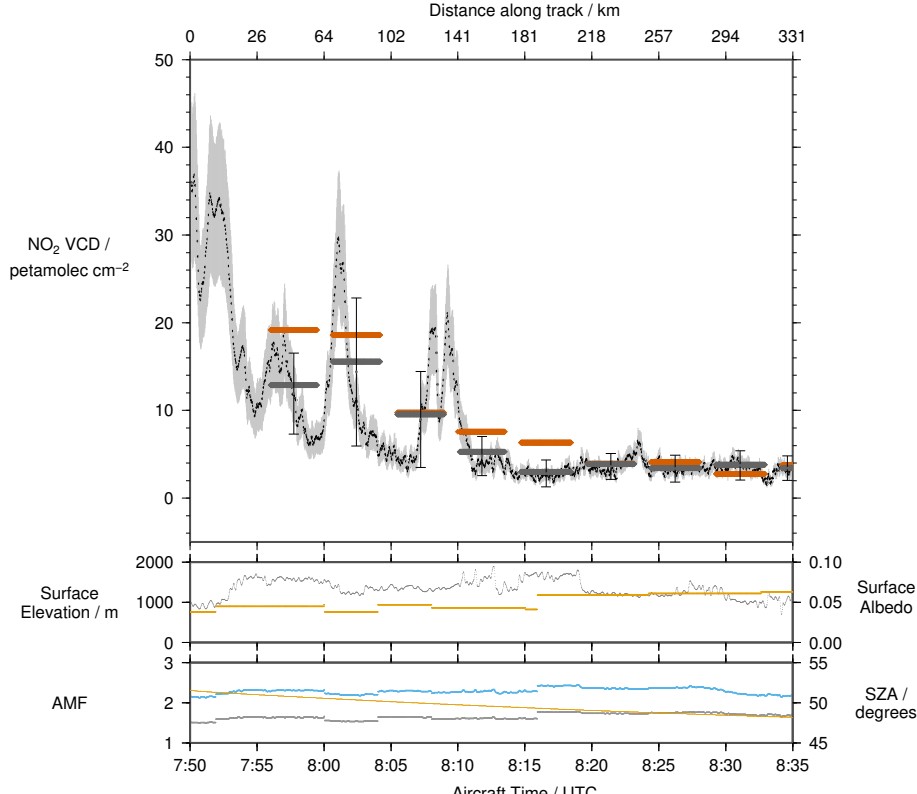

**Figure 17.** A timeseries of the airborne DOAS NO$_2$ vertical column densities on 14 August 2007 , and SCIAMACHY at aircraft nadir (orange). Aircraft measurements averaged over the length of each SCIAMACHY pixel are shown in grey. Sub-plots are as described for Fig. 10

Heue, K.-P., Wagner, T., Broccardo, S. P., Walter, D., Piketh, S. J., Ross, K. E., Beirle, S., and Platt, U.: Direct observation of two dimensional trace gas distributions with an airborne Imaging DOAS instrument, Atmospheric Chemistry and Physics, 8, 6707–6717, doi:10.5194/acp-8-6707-2008, http://www.atmos-chem-phys.net/8/6707/2008/, 2008.

Hobbs, P. V.: Clean air slots amid dense atmospheric pollution in southern Africa, Journal of Geophysical Research, 108, 1–8,
5   doi:10.1029/2002JD002156, http://www.agu.org/pubs/crossref/2003/2002JD002156.shtml, 2003.

Holben, B. N., Tanre, D., Smirnov, A., Eck, T. F., Slutsker, I., Abuhassan, N., Newcomb, W. W., Schafer, J. S., Chatenet, B., Lavenu, F., Kaufman, Y. J., Castle, J. V., Setzer, A., Markham, B., Clark, D., Frouin, R., Halthore, R., Karneli, A., O'Neill, N. T., Pietras, C., Pinker, R. T., Voss, K., and Zibordi, G.: An emerging ground-based aerosol climatology : Aerosol optical depth from AERONET, Journal of Geophysical Research, 106, 12 067–12 097, 2001.

10   Irie, H., Kanaya, Y., Akimoto, H., Tanimoto, H., Wang, Z., Gleason, J. F., and Bucsela, E. J.: Validation of OMI tropospheric NO 2 column data using MAX-DOAS measurements deep inside the North China Plain in June 2006 : Mount Tai Experiment 2006, Atmospheric Chemistry and Physics, pp. 6577–6586, 2008.

Kanaya, Y., Irie, H., Takashima, H., Iwabuchi, H., Akimoto, H., Sudo, K., Gu, M., Chong, J., and Kim, Y. J.: Long-term MAX-DOAS network observations of NO 2 in Russia and Asia ( MADRAS ) during the period 2007 – 2012 : instrumentation, elucidation of climatology ,

and comparisons with OMI satellite observations and global model simulations, Atmospheric Chemistry and Physics, pp. 7909–7927, doi:10.5194/acp-14-7909-2014, 2014.

Kleipool, Q. L., Dobber, M. R., de Haan, J. F., and Levelt, P. F.: Earth surface reflectance climatology from 3 years of OMI data, Journal of Geophysical Research: Atmospheres, 113, 1–22, doi:10.1029/2008JD010290, 2008.

Kokhanovsky, A. A. and Rozanov, V. V.: Retrieval of NO 2 vertical columns under cloudy conditions : A sensitivity study based on SCIATRAN calculations, Atmospheric Research, 93, 695–699, doi:10.1016/j.atmosres.2009.01.022, http://dx.doi.org/10.1016/j.atmosres.2009.01.022, 2009.

Kraus, S. G.: DOASIS: A Framework Design for DOAS, Ph.D. thesis, Mannheim, 2006.

Laakso, L., Vakkari, V., Virkkula, a., Laakso, H., Backman, J., Kulmala, M., Beukes, J. P., van Zyl, P. G., Tiitta, P., Josipovic, M., Pien-
aar, J. J., Chiloane, K., Gilardoni, S., Vignati, E., Wiedensohler, a., Tuch, T., Birmili, W., Piketh, S. J., Collett, K., Fourie, G. D., Komppula, M., Lihavainen, H., de Leeuw, G., and Kerminen, V.-M.: South African EUCAARI measurements: seasonal variation of trace gases and aerosol optical properties, Atmospheric Chemistry and Physics, 12, 1847–1864, doi:10.5194/acp-12-1847-2012, http://www.atmos-chem-phys.net/12/1847/2012/, 2012.

Leitão, J., Richter, A., Vrekoussis, M., Kokhanovsky, A., Zhang, Q. J., Beekmann, M., and Burrows, J.: On the improvement of NO2 satellite
retrievals – aerosol impact on the airmass factors, Atmospheric Measurement Techniques, 3, 475–493, doi:10.5194/amt-3-475-2010, 2010.

Levelt, P. F., Oord, G. H. J. V. D., Dobber, M. R., Mälkki, A., Visser, H., Vries, J. D., Stammes, P., Lundell, J. O. V., and Saari, H.: The Ozone Monitoring Instrument, IEEE Transactions on Geoscience and Remote Sensing, 44, 1093–1101, 2006.

Lin, J.-T., Martin, R. V., Boersma, K. F., Sneep, M., Stammes, P., Spurr, R., Wang, P., Van Roozendael, M., Clémer, K., and Irie, H.: Retrieving tropospheric nitrogen dioxide from the Ozone Monitoring Instrument: effects of aerosols, surface reflectance
anisotropy, and vertical profile of nitrogen dioxide, Atmospheric Chemistry and Physics, 14, 1441–1461, doi:10.5194/acp-14-1441-2014, http://www.atmos-chem-phys.net/14/1441/2014/, 2014.

Liu, Y. and Daum, P. H.: THE EFFECT OF REFRACTIVE INDEX ON SIZE DISTRIBUTIONS AND LIGHT SCATTERING COEFFICIENTS DERIVED FROM OPTICAL PARTICLE COUNTERS, Journal of Aerosol Science, 31, 945–957, 2000.

Lourens, A., Butler, T., Beukes, J. P., van Zyl, P. G., Beirle, S., Wagner, T., Heue, K.-P., Pienaar, J. J., Fourie, G. D., and Lawrence, M. G.:
Re-evaluating the NO 2 hotspot over the South African Highveld, South African Journal of Science, 108, 1–6, 2012.

Maenhaut, W., Salma, I., and Cafrneyer, J.: Regional atmospheric aerosol composition and sources in the eastern Transvaal, South Africa, and impact of biomass burning, Journal of Geophysical Research, 101, 23 613–23 650, 1996.

Magi, B. I., Hobbs, P. V., Schmid, B., and Redemann, J.: Vertical profiles of light scattering, light absorption, and single scattering albedo during the dry, biomass burning season in southern Africa and comparisons of in situ and remote sensing measurements of aerosol optical
depths, Journal of Geophysical Research, 108, doi:10.1029/2002JD002361, http://www.agu.org/pubs/crossref/2003/2002JD002361.shtml, 2003.

Martin, R. V., Chance, K. V., Jacob, D. J., Kurosu, T. P., Spurr, R. J. D., Bucsela, E. J., Gleason, J. F., Palmer, P. I., Bey, I., Fiore, A. M., Li, Q., Yantosca, R. M., and Koelemeijer, R. B.: An improved retrieval of tropospheric nitrogen dioxide from GOME, Journal of Geophysical Research, 107, doi:10.1029/2001JD001027, http://www.agu.org/pubs/crossref/2002/2001JD001027.shtml, 2002.

McLinden, C. a., Fioletov, V., Boersma, K. F., Krotkov, N. A., Sioris, C. E., Veefkind, J. P., and Yang, K.: Air quality over the Canadian oil sands: A first assessment using satellite observations, Geophysical Research Letters, 39, n/a–n/a, doi:10.1029/2011GL050273, http://doi.wiley.com/10.1029/2011GL050273, 2012.

Mclinden, C. A., Fioletov, V., Shephard, M. W., Krotkov, N., Li, C., Martin, R. V., Moran, M. D., and Joiner, J.: Space-based detection of missing sulfur dioxide sources of global air pollution, Nature Geoscience, pp. 1–7, doi:10.1038/NGEO2724, 2016.

Platt, U. and Stutz, J.: Differential Optical Absorption Spectroscopy, Springer-Verlag, Berlin, Heidelberg, 1st edn., 2008.

Rhodes, B.: pyEphem Home Page, http://rhodesmill.org/pyephem/, 2015.

Richter, A., Burrows, J. P., Nüss, H., Granier, C., and Niemeier, U.: Increase in tropospheric nitrogen dioxide over China observed from space., Nature, 437, 129–32, doi:10.1038/nature04092, http://www.ncbi.nlm.nih.gov/pubmed/16136141, 2005.

Rosenberg, P. D., Dean, A. R., Williams, P. I., Dorsey, J. R., Minikin, A., Pickering, M. A., and Petzold, A.: Particle sizing calibration with refractive index correction for light scattering optical particle counters and impacts upon PCASP and CDP data collected during the Fennec campaign, Atmospheric Measurement Techniques, 5, 1147–1163, doi:10.5194/amt-5-1147-2012, 2012.

Rothman, L., Rinsland, C., Goldman, a., Massie, S., Edwards, D., Flaud, J.-M., Perrin, a., Camy-Peyret, C., Dana, V., Mandin, J.-Y., Schroeder, J., Mccann, a., Gamache, R., Wattson, R., Yoshino, K., Chance, K., Jucks, K., Brown, L., Nemtchinov, V., and Varanasi, P.: the Hitran Molecular Spectroscopic Database and Hawks (Hitran Atmospheric Workstation): 1996 Edition, Journal of Quantitative Spectroscopy and Radiative Transfer, 60, 665–710, doi:10.1016/S0022-4073(98)00078-8, http://linkinghub.elsevier.com/retrieve/pii/S0022407398000788, 1998.

Rozanov, V. V. and Rozanov, A. V.: Differential optical absorption spectroscopy (DOAS) and air mass factor concept for a multiply scattering vertically inhomogeneous medium: theoretical consideration, Atmospheric Measurement Techniques, 3, 751–780, doi:10.5194/amt-3-751-2010, http://www.atmos-meas-tech.net/3/751/2010/, 2010.

Rozanov, V. V., Rozanov, A. V., Kokhanovsky, A. A., and Burrows, J. P.: Radiative transfer through terrestrial atmosphere and ocean: Software package SCIATRAN, Journal of Quantitative Spectroscopy and Radiative Transfer, 133, 13–71, doi:10.1016/j.jqsrt.2013.07.004, http://dx.doi.org/10.1016/j.jqsrt.2013.07.004, 2014.

Stein, A. F., Draxler, R. R., Rolph, G. D., Stunder, B. J. B., Cohen, M. D., and Ngan, F.: Noaa's hysplit atmospheric transport and dispersion modeling system, Bulletin of the American Meteorological Society, 96, 2059–2077, doi:10.1175/BAMS-D-14-00110.1, 2015.

Streets, D. G., Canty, T., Carmichael, G. R., Foy, B. D., Dickerson, R. R., Duncan, B. N., Edwards, D. P., Haynes, J. A., Henze, D. K., Houyoux, M. R., Jacob, D. J., Krotkov, N. A., Lamsal, L. N., Liu, Y., Lu, Z., Martin, R. V., Gabriele, G. P., Pinder, R. W., Salawitch, R. J., and Wecht, K. J.: Emissions estimation from satellite retrievals : A review of current capability, Atmospheric Environment, 77, 1011–1042, doi:10.1016/j.atmosenv.2013.05.051, 2013.

Swap, R. J. and Tyson, P. D.: Stable discontinuities as determinants of the vertical distribution of aerosols and trace gases in the atmosphere, South African Journal of Science, 95, 63–71, 1999.

Toenges-Schuller, N., Stein, O., Rohrer, F., Wahner, a., Richter, A., Burrows, J. P., Beirle, S., Wagner, T., Platt, U., and Elvidge, C. D.: Global distribution pattern of anthropogenic nitrogen oxide emissions: Correlation analysis of satellite measurements and model calculations, Journal of Geophysical Research, 111, D05 312, doi:10.1029/2005JD006068, http://doi.wiley.com/10.1029/2005JD006068http://www.agu.org/pubs/crossref/2006/2005JD006068.shtml, 2006.

van der A, R. J., Eskes, H. J., Boersma, K. F., Noije, T. P. C. V., Roozendael, M. V., De Smedt, I., Peters, D. H. M. U., Meijer, E. W., van Noije, T. P. C., Van Roozendael, M., De Smedt, I., Peters, D. H. M. U., and Meijer, E. W.: Trends, seasonal variability and dominant NO x source derived from a ten year record of NO 2 measured from space, Journal of Geophysical Research, 113, 1–12, doi:10.1029/2007JD009021, http://www.agu.org/pubs/crossref/2008/2007JD009021.shtml, 2008.

Vandaele, A. C., Hermans, C., Simon, P. C., Carleer, M., Colin, R., Fally, S., Merienne, M. F., Jenouvrier, A., and Coquart, B.: MEASURE-MENTS OF THE NO2 ABSORPTION CROSS-SECTION FROM 42 000 cm-1 TO 10 000 cm-1 ( 238-1000 nm) AT 220 K AND 294 K, Journal of Quantitative Spectroscopy and Radiative Transfer, 59, 171–184, 1998.

Veefkind, J. P., Aben, I., McMullan, K., Förster, H., de Vries, J., Otter, G., Claas, J., Eskes, H., de Haan, J., Kleipool, Q., van Weele, M., Hasekamp, O., Hoogeveen, R., Landgraf, J., Snel, R., Tol, P., Ingmann, P., Voors, R., Kruizinga, B., Vink, R., Visser, H., and Levelt, P. F.: TROPOMI on the ESA Sentinel-5 Precursor: A GMES mission for global observations of the atmospheric composition for climate, air quality and ozone layer applications, Remote Sensing of Environment, 120, 70–83, doi:10.1016/j.rse.2011.09.027, http://linkinghub.elsevier.com/retrieve/pii/S0034425712000661, 2012.