# Peer review of "Intra-pixel variability in satellite tropospheric $NO_2$ column densities derived from simultaneous space-borne and airborne observations over the South African Highveld"

_Atmospheric Measurement Techniques, 2016_

## Referee Comment (RC1) · Anonymous Referee #1 · 6 Mar 2017

This paper presents results from an airborne field campaign in 2007 over the Highveld region of South Africa. The authors compare NO2 measurements from the iDOAS airborne instrument with NO2 columns from the OMI and SCIAMACHY satellite instruments taken from the www.temis.nl data archive. On the whole, the paper is very well written and organized, and easy to read. The results and conclusions are not very surprising (OMI and SCIAMACHY, with lower spatial resolution observations than iDOAS, are unable to resolve the high values in a plume), but the paper is a nice description of several case studies and a record of the campaign, and provides some interesting context for those looking at satellite measurements over the Highveld region. I would

recommend it be published in AMT after addressing several minor comments below.

The title could be the title of many papers, both previous and upcoming, but this is a fairly narrow study of one field campaign in one location involving one airborne instrument and 4 flights. The data shown is even averaged across-track, so the true variability is not really easy to assess in the paper. There aren't really any very general conclusions to draw from the paper, or general recommendations on how to deal with these descrepancies, that really warrant such a general title. Maybe add something like: "Results from the X campaign over the South African Highveld", or mention iDOAS, SCIA, OMI etc for a more descriptive title. I apologize – this is something I should have pointed out in my initial quick review of the paper.

I see there are some figures in the Heue et al. instrument paper showing a sample swath of NO2 observations below the instrument, but I think some kind of figure showing the original NO2 measurements and their spatial extent and resolution could help put these observations in context in this paper. Also, it is very hard to tell from the figures of OMI/SCIA and iDOAS measurements (Fig 6 etc) that this is a 2-D type of observation. Are these plots only showing the average value every 1.2 km or is there 2-D resolved NO2 in the plots? It's hard to tell and I don't remember seeing it in the text (maybe add to figure captions). Also, it's quite hard to see the colors in some of the plots along the flight track. I'm worried when the figures become the size of a single column it may be even more difficult to see – just ensure at the smaller size that the NO2 colours are indeed visible. You mention the swath width but it is quite late in the paper. I think this could help describe observations if it were mentioned earlier in the paper with the discussion of spatial resolution.

Page 2, Line 32: The slant density is the integral of the path length times the number density of that absorber, not the concentration (which describes the number density absorber as a fraction of the total air density).

Page 3, Line 4: "to a first approximation, is slanted" is a bit confusing. Do you mean

because of geometry?

Page 3, Line 12: The analysis of the NO2 slant column is skimmed over without really any detail. I realize there is another paper describing this process, but could you say a few words about what other absorbers and parameters are fit, as well as individual fitting uncertainties from noise or systematic uncertainties? Also, not much info on instrument. What is SNR, are these from spectra that have been co-added spatially, what is the size of the CCD array (pixels), spectral resolution, spectral sampling etc?What is used for a reference spectrum?

Page 4, Line 2: Can you expand just briefly on why a photolytic converter is desirable? Also, why do you present NOy and not NO2?

Section 2: Subheadings would increase the readability of this section. For example: "iDOAS NO2", "In situ measurements", "Satellite observations" etc.

Page 4, Line 16: This only the best case at nadir. The sides of the OMI swath are much larger.

Page 4, Line 17: Can you give uncertainties in satellite VCD's? These can be quite large.

Page 4, Line 20: I got confused here as on initial reading it sounded like the TM4NO2A was OMI data but with SCIA stratospheric slant columns as strat columns had just been mentioned.

Page 4, Line 26: Suggest mentioning swath width here and how many across track pixels there are here.

Page 5, Line 12: Why do you average to 1.2 km? If purpose is to examine intra-pixel variability, how much cross-track and along-track information are you losing? Is this done to reduce error from noise?

Page 6, Line 2: Are there only 8 profiles total and what are locations? Maybe mention

here to put in context. This intro to the section is a bit confusing as it presents the conclusion all of a sudden without referencing the data/figures. Maybe add an introductory sentence to ease into the analysis.

Page 6: Is the representation of some profiles as exponential valid in this region? Do you have any surface observations, or model profiles to check against? I realize this is done for contraining the error more than anything, so probably doesn't make a big difference, but I'm just curious.

Figure 2 and 3: Can you specify surface altitude here or show on figure? Is it the bottom of the y-axis?

Figure 4 and relevant text: What do you do for NO2 profile in stratosphere in the model? Does NO2 in the stratosphere contribute to the AMF or do you assume it cancels perfectly with the reference spectrum?

Section 5: Could increase readability with subsection headings here.

Page 12, Line 11: Clearly the AMF changes drastically with the surface albedo. You are using your calculated AMF values to set bounds on the AMF error. How is the uncertainty in the AMF from uncertainty in the albedo determined (it's going to be high, with such a low resolution OMLER product)? Why not use MODIS albedo, or MODIS BRDF for an even better represention of the surface for high resolution observations from the aircraft?

Page 15, Line 15: I found this confusing. What is your reference? Do you have remote ocean measurements from the aircraft?

Page 14, Line 14: Could the same effect be achieved by putting the iDOAS on an aircraft that flies higher?

Page 14, 19: "Underestimates" the peak only. What it's actually probably doing is just averaging out everything in the field of view (so you could equally say "overestimates" the background).

Page 13, Line 5: Not sure why you have to use two fitting schemes. Does that tell us anything?

Page 15: You make a few comments about plume age, source etc. I noticed you used HYSPLIT earlier in the paper. Can it tell you anything about these specific cases?

Page 16, Line 19: Can you remind us of SCIA overpass time here? Also Figure 13 caption reads a bit like the times are for the satellite observations (I'm guessing it didn't take 50 minutes to fly over the region!)

Page 17, Line 6: Not sure you can draw any conclusions about SCIAMACHY vs OMI at all here. There is a very limited amount of SCIA data at high NO2 values. Obviously your slopes are very different on different days with OMI as well.

Fig 6 and similar: Can you specify that these are 1.2 km averages in caption or in text (which I'm assuming they must be?)

Figure 6: I can't tell which is sub-aircraft pixel as it looks like flight was right down the border of two cross-track positions. Can you clarify this in text?

Figure 7 and similar, and relevant text: The average isn't technically over the "area" of the OMI pixel, which might have a 13x24 km2 size. Clarify.

Figure 7 and similar: Specify colors for elevation/albedo subplot.

Figure 8: Specify which are OMI and which are SCIA observations (maybe in legend?)

---

## Referee Comment (RC2) · Anonymous Referee #3 · 23 May 2017

This manuscript presents a study on the comparison of airborne versus satellite tropospheric NO2 retrievals over the South African Highveld region. The study focuses mostly on the interpretation of four research flights performed in August 2007 that coincide with measurements from OMI and SCIAMACHY.

This work is highly relevant for two reasons: first because of the mismatch reported in the literature between tropospheric NO2 columns derived from bottom-up inventories and those from satellite retrievals in this region (satellite observations provide higher values). Before one may conclude that local emission inventories need to be adapted,

satellite retrievals should be validated locally. Secondly, this study has a global relevance in the sense that it gives valuable insights with respect to the horizontal variability of tropospheric NO2 columns when observed at high spatial resolution. The satellite instruments OMI and SCIAMACHY are known to have a moderate spatial resolution and it is not well known how this resolution compares to the typical scale of spatial variabilities in the urban tropospheric NO2 column field. This information is relevant for many related studies where satellite retrievals are used (e.g. in order to derive top-down emission inventories) or being validated (e.g. comparison with MAX-DOAS).

The manuscript is generally well written and addresses a relevant topic. However, to my opinion several aspects deserve more attention before publication in AMT.

1 / Four research flights are analysed and discussed in quite some detail, but the interpretation of differences between airborne and satellite retrievals could go more into depth. Based on the present manuscript, the reader might get the impression that systematic differences in tropospheric NO2 columns between satellite and iDOAS can be explained solely (or largely) by horizontal variability in the tropospheric NO2 columns on a scale that is smaller than the typical size of satellite pixels. What could be particularly relevant is to investigate further the potential impact of profile shape assumptions for NO2 and aerosols in explaining the difference between satellite and airborne measurements over the most polluted regions. Close to major point sources one may expect not only to find locally quite extreme tropospheric NO2 column abundances, but at the same locations also the NO2 profile shape may deviate considerably from other places further away from the main sources. In this context, it may be relevant to distinguish explicitly four profiles: the true profile at the spatial resolution of the aircraft measurements (P_true_air), the profile used in the airborne retrieval (P_prior_air), the true profile at the resolution of the satellite measurements (P_true_sat) and the profile used in the satellite retrieval (P_prior_sat). Differences in tropospheric NO2 column retrievals (space borne versus airborne) cannot be interpreted without taking into including these four profile shapes in the discussion: how much do the authors think

P_true_air can deviate from P_prior_air close to the main sources (same for P_true_sat and P_prior_sat). Furthermore the AMF is not only affected by the (different) profile shapes, but also by the block-AMFs, and these are not identical for the satellite and the airborne point of view. This should be taken into account as well.

Despite the length of this comment, I would suggest to add just one or two paragraphs addressing this point and providing some first order estimates. It could for instance be enlightening to the reader if the impact of making wrong profile shape assumptions is worked out for one hypothetical scenario. For instance (it is up to the authors to deviate from this concrete suggestion): scale height for P_true_air is 0.2 km (e.g. close to strong isolated source); scale height for P_true_sat is 0.4 km (averaged over a larger region the true profile is less dominated by the local source); scale height for P_prior_air is 0.6 km (this number is used in present study); scale height for P_prior_sat is taken from profile used in DOMINOv2 product over this region. Block_AMFs should be applied for a representative SZA and surface reflectance. When combined, this information should provide the reader with a first order quantitative estimate of local AMF fluctuations near a strong plume: to what extent can this explain the discrepancy between the satellite and airborne retrieval? Or perhaps it is concluded that - when taking this effect into account - the observed discrepancy increases even further.

2 / Although aerosols are not entirely neglected in this study, they receive little attention considering the fact that for all four flights - each covering distances of hundreds of kilometers - just one fixed value is assumed for the AOT. It is quite remarkable that the uncertainty range of the AMF is derived using a look-up table that does include variability of the single scattering albedo, but not of the AOT. Over a region where the variability in NO2 is so large, it is almost unthinkable that the AOT can be approximated with a single value. To some extent the same argumentation as given above (in the vicinity of a strong pollution source the NO2 profile shapes may show considerable spatial variability) can be given here as well: in the same region the AOT may show a substantial variability (although probably less extreme than for NO2). In my opinion

this point should at least be mentioned. It would be even better to find satellite AOT data (e.g. from MODIS) for the days of the research flights to provide more insight into this relevant parameter.

3 / In the manuscript the discrepancies found between iDOAS and OMI (SCIAMACHY) are not compared to results from other validation studies, e.g. where OMI retrievals are compared to MAX-DOAS observations. In the last years many of such studies were done, with MAX-DOAS instruments either in rural or in urban regions. It would be valuable to link the findings of this study to findings in such inter-comparisons.

4 / On section 2: please provide some more details on the iDOAS observations. For instance: the field of view, number of ground pixels in across track direction.

5 / I am missing a formula that describes how VCD's are derived precisely from the (differential) slant column measurements. In my opinion, this should be described in a more detail, although it has already been described elsewhere in full detail.

6 / The statements in Sect. 3 are quite general. The words 'usually' (p.6,l.3) and 'frequently' (p.6,l.8) suggest a large number of profiles that are measured. However, these are not shown. Furthermore, it is not clear if the profiles that are measured are representative for the plume or for more remote regions (see also the first comment).

7 / Figure 5 could be better readable if a grid was plotted on the left and right side of each cube. Furthermore it could be beneficial to use colours instead of different line styles and to provide a legend.

8 / P.15, l.24-25: ". . .indicating that . . . 9 August". The terminology 'aged' versus 'young' might cause confusion, as some readers might wrongly think of 'photochemical aging'. It might be that what is here called an 'aged plume' is actually a region where the $NO_2$ profile is less shallow than for a 'young plume', and more in line with the prior $NO_2$ profile shape used for the OMI and/or iDOAS retrievals (see also the first comment). If this is the case, then one cannot say that OMI would be limited in its ability to capture

the higher NO2 gradient in the young plume because it is 'young'; for instance it could be more appropriate to say that the AMF derived using the prior profile shape used in the DOMINO product better matches the profile shape of an aged plume than the profile shape of a young plume. Please comment on this.
* * *

---

## Author Comment (AC1) · 20 Jun 2017

**AMT-2016-366: Responses to Anonymous Reviewer 1**

**Stephen Broccardo**

**June 20, 2017**

Thank you for the positive review and comments.

The title of the paper has been changed to reflect the fact that the results are specifically from the Highveld.

In response to the specific comments:

***Page 2, Line 32:*** *The slant density is the integral of the path length times the number density of that absorber, not the concentration (which describes the number density absorber as a fraction of the total air density).*

**Response and action**

Agreed. Edited p2. line 32 and p3. line 4: changed the word "concentration" to "molecular number density".

*Page 3, Line 4: "to a first approximation, is slanted" is a bit confusing. Do you mean because of geometry?*

**Response and action**

Yes, a geometric first approximation is what is meant. Added the word "geometric" to the sentence.

***Page 3, Line 12:*** *The analysis of the $NO_2$ slant column is skimmed over without really any detail. I realize there is another paper describing this process, but could you say a few words about what other absorbers and parameters are fit, as well as individual fitting uncertainties from noise or systematic uncertainties? Also, not much info on instrument. What is SNR, are these from spectra that have been co-added spatially, what is the size of the CCD array (pixels), spectral resolution, spectral sampling etc? What is used for a reference spectrum?*

***Page 4, Line 2:*** *Can you expand just briefly on why a photolytic converter is desirable? Also, why do you present NOy and not $NO_2$?*

**Response and Action**

In response to these two questions, the description of the measurements has been expanded. The paragraphs now read:

A DOAS instrument based on an Acton 300i imaging spectrograph employing a pushbroom viewing geometry, where each line of pixels across the instrument's swath is captured simulaneously on an Andor DU-420BU CCD, was fitted into the Aerocommander 690A. This CCD has 255 pixels in the across-track dimension and 1024 pixels in the spectral direction. The temperature of the spectrograph was kept stable at 30°C using a thermostatic heater in an insulated box, and the CCD temperature was set at -20°C using its own in-built thermo-electric cooler. Eight spectra were co-added into 32 across-track pixels, each with an across-track footprint of approximately 70m, assuming a flight altitude of 4500m above the ground. This was done in order to make optimum use of the optical resolution of the instrument. Along-track resolution is determined by the aircraft speed and the integration time on the instrument, which was adjusted automatically in-flight to avoid saturation of the CCD, and is generally about 100m (Heue et al, 2008). In the present study only the nadir pixel of the iDOAS is used.

Slant-column densities were retrieved using the WinDOAS software package. Absorption cross-sections for $NO_2$ (Vandaele et al, 1998), ozone (Burrows et al, 1999), water vapour (Rothman et al, 1998), $O_4$ (Greenblatt et al, 1990) were fitted across a spectral range of 432nm to 464nm. The Ring effect was accounted for using a appropriate cross-section calculated using the DOASIS software (Kraus, 2006). A reference spectrum was chosen from an appropriate location along the flight track far from known sources implying that slant-column densities from WinDOAS are in fact differential slant column densites. Satellite retrievals use a similar technique, using a measurement over remote ocean areas as an approximation of zero-$NO_2$. We adjust our slant-column densities using an offset in order to bring the vertical column densities from the iDOAS into line with the appropriate satellite measurement (either OMI or SCIAMACHY) in background areas of our flight track.

In addition to the imaging DOAS (iDOAS) instrument, the aircraft carried a Particle Measurement Systems Passive Cavity Aerosol Spectrometer Probe 100X (PCASP), operated with the pre-heater switched on; and a Thermo Scientific 42i chemiluminescence instrument with a molybdenum converter in the cabin, plumbed into the aircraft's scientific-air inlet in order to measure in-situ $NO_y$. In such instruments the converter converts $NO_2$ to NO, which is then measured by chemiluminescence; however a molybedenum converter also converts other nitrogen species. This can be avoided using a photolytic converter, however an instrument with a photolytic converter to measure $NO_2$ was not within the project's budget. The aircraft is fitted with a Rosemount ambient temperature sensor, and a separate pitot-static

system for measurement and logging of static and dynamic pressure. The humidity sensor fitted to the aircraft did not function during this campaign. The aircraft's data aquisition system also logged parameters from a GPS (Global Positioning System) receiver.

*Section 2: Subheadings would increase the readability of this section. For example: "iDOAS NO$_2$", "In situ measurements", "Satellite observations" etc.*

**Response and action**

Subheadings have been added, and some of the paragraphs re-arranged to be under the relevant heading.

*Page 4, Line 16: This only the best case at nadir. The sides of the OMI swath are much larger.*

**Response and action**

The broadening of the OMI ground-pixel size towards the edges of the swath has been clarified, with the addition of the sentence "OMI pixels broaden in the across track direction as the viewing angle moves away from nadir".

*Page 4, Line 17: Can you give uncertainties in satellite VCD's? These can be quite large.*

**Response**

Uncertainties in the satellite VCD's are provided in the data files from the satellite retrievals, and could be shown. However, the figures in the present paper are already quite busy, and the focus is on AMF uncertainties in the iDOAS and satellite measurements (which, as shown in the response to reviewer 3's comments, we feel may be more uncertain than previously thought) and variability within each satellite pixel. We feel that trying to show too much will detract from this focus.

*Page 4, Line 20: I got confused here as on initial reading it sounded like the TM4NO2A was OMI data but with SCIAMACHY stratospheric slant columns as strat columns had just been mentioned.*

**Action**

The sentence has been changed to: "The TM4NO2A product is a product using slant column measurements from the SCIAMACHY satellite instrument and a similar scheme using model profiles and stratospheric columns from the TM4 model."

*Page 4, Line 26: Suggest mentioning swath width here and how many across track pixels there are here.*

**Action**

This has been added to the improved description of the DOAS measurements earlier in the section.

**Page 5, Line 12:** *Why do you average to 1.2 km? If purpose is to examine intra-pixel variability, how much cross-track and along-track information are you losing? Is this done to reduce error from noise?*

**Response**

This is in fact a time-based average (10 second moving average), which at the flight speed works out to approximately 1.2km. The iDOAS has 32 across-track pixels, we use only pixel 15 (counting from 0) in this paper.

Along-track averaging is used to reduce noise, for example an earlier version of Figure 7, without the averaging, and using a fixed AMF rather than attempting to quantify iDOAS AMF uncertainty is shown below.

[Figure]

Figure 1: Airborne iDOAS measurements at full resolution (i.e. approx. 80m) on 9 August 2007 compared with OMI DOMINO V2 at aircraft nadir (orange) and one pixel upwind of (blue). A fixed AMF of 1.6 is used for the aircraft measurements in this figure.

**Action**

In the section on comparing DOAS measurements to the satellite (p5, line 9, which has now been moved under the appropriate sub-heading) the following has been added: "...the first is to average 80m-resolution nadir iDOAS measurements using a ten-second moving average in order to smooth out

fine-spatial-scale variations and make a comparison with the much larger satellite pixels." and "The second approach is to calculate the mean and standard deviation of all nadir iDOAS measurements . . . "

***Page 6, Line 2:*** *Are there only 8 profiles total and what are locations? Maybe mention here to put in context. This intro to the section is a bit confusing as it presents the conclusion all of a sudden without referencing the data/figures. Maybe add an introductory sentence to ease into the analysis.*

**Response and action**

Yes there are only 8 profiles. An introductory sentence has been added: "During each of the flights, a vertical profile measurement was performed before and after the satellite-tracking portion of the flight."

***Page 6:*** *Is the representation of some profiles as exponential valid in this region? Do you have any surface observations, or model profiles to check against? I realize this is done for contraining the error more than anything, so probably doesn't make a big difference, but I'm just curious.*

**Response**

We don't have any model profiles that we feel will be helpful. The state of modelling in South Africa is poor, mostly relevant to regulatory compliance. There are some other unpublished vertical profiles from other campaigns, but we chose to use only those from the 2007 iDOAS campaign. Near to surface sources, and at the spatial scale of the iDOAS, an exponential profile seems more appropriate. This is difficult to confirm, even with an aircraft profile measurement, since aircraft will typically climb and descend in a racecourse pattern, with straight portions of two minutes' flying between 180° turns, amounting to almost 10km of flying. Perhaps an ex-military pilot would be willing to fly more aggressively to make the horizontal extent of vertical profile measurements smaller. This would also require greater coordination with air-traffic control, who might find this sort of maneuvering unusual. A vertical profile climatology would be useful over the Highveld and in fact most regions of the world.

***Figure 2 and 3:*** *Can you specify surface altitude here or show on figure? Is it the bottom of the y-axis?*

**Response and action**

Surface elevation at Richards Bay is sea level, and at Nelspruit the aircraft landed to refuel, so the profile is measured down to the surface. This information has been added to the captions and the text.

***Figure 4 and relevant text:*** *What do you do for $NO_2$ profile in stratosphere in the model? Does $NO_2$ in the stratosphere contribute to the AMF or do you assume it cancels perfectly with the reference spectrum?*

**Response**

We assume that the change in stratospheric AMF during the course of a flight is small, and hence we assume that the stratospheric column cancels with the reference spectrum.

*Section 5: Could increase readability with subsection headings here.*

Subheadings have been added to separate the descriptions of flight on different days.

*Page 12, Line 11: Clearly the AMF changes drastically with the surface albedo. You are using your calculated AMF values to set bounds on the AMF error. How is the uncertainty in the AMF from uncertainty in the albedo determined (it's going to be high, with such a low resolution OMLER product)? Why not use MODIS albedo, or MODIS BRDF for an even better represention of the surface for high resolution observations from the aircraft?*

**Response**

Certainly, using a higher resolution albedo or BRDF would improve our calculation of the AMF, since surface properties have a large influence on the AMF. This would be the logical next step in sophistication of the radiative transfer modelling. It would appear that the uncertainty in the AMF from the profile shape is more important, and more difficult to quantify.

*Page 15, Line 15: I found this confusing. What is your reference? Do you have remote ocean measurements from the aircraft?*

**Response**

Reference spectra for our retrieval are chosen from background regions of the flight. Since background regions over land will have a higher column density than remote maritime regions measured by the satellites, we assume that the satellites' measurement of background regions over land are "correct" and we shift the aircraft measurements to match the satellite. This seems reasonable, since background regions have shallow horizontal gradients and the errors in the satellite measurements we describe will be small. This procedure does highlight the problem with using one nadir-viewing scattered-light instrument to validate another. Ideally some other independent measure of vertical column density, with less dependence on *a-priori* profile shape should be used as a reference for both the aircraft and satellite measurements.

*Page 14, Line 14: Could the same effect be achieved by putting the iDOAS on an aircraft that flies higher?*

Yes, a wider swath could be achieved by flying higher.

***Page 14, 19:*** *"Underestimates" the peak only. What it's actually probably doing is just averaging out everything in the field of view (so you could equally say "overestimates the background").*

The regression line is fitted through iDOAS data that is averaged (along track) to the resolution of OMI, hence we are not comparing the peak iDOAS to OMI but rather a 1-dimensional spatial average.

***Page 13, Line 5:*** *Not sure why you have to use two fitting schemes. Does that tell us anything?*

**Response**

The two fitting schemes attempt to demonstrate that the background measurements by the iDOAS and satellites are a better match than measurements close to sources. However, the reviewr makes a good point: it doesn't really tell us much.

**Action**

The inset figure has been removed.

***Page 15:*** *You make a few comments about plume age, source etc. I noticed you used HYSPLIT earlier in the paper. Can it tell you anything about these specific cases?*

In cases where HYSPLIT proved to be useful for plume age estimates, this has been added to the discussion.

***Page 16, Line 19:*** *Can you remind us of SCIAMACHY overpass time here? Also Figure 13 caption reads a bit like the times are for the satellite observations (I'm guessing it didn't take 50 minutes to fly over the region!)*

SCIAMACHY overpass times are shown in Figure 13 in each SCIAMACHY pixel. The approximate overpass time has been added to the caption.

***Page 17, Line 6:*** *Not sure you can draw any conclusions about SCIAMACHY vs OMI at all here. There is a very limited amount of SCIAMACHY data at high $NO_2$ values. Obviously your slopes are very different on different days with OMI as well.*

True. We have edited the text appropriately to indicate that no real conclusion can be drawn.

***Fig 6 and similar:*** *Can you specify that these are 1.2 km averages in caption or in text (which I'm assuming they must be?)*

Yes these are 1.2km averages. This can be added to the caption.

***Figure 6:*** *I can't tell which is sub-aircraft pixel as it looks like flight was right down the border of two cross-track positions. Can you clarify this in*

*text?*

Yes, the flight track does cross from one OMI-row to the next. This has been clarified in the description of the figure.

**Figure 7 and similar,** *and relevant text: The average isn't technically over the "area" of the OMI pixel, which might have a 13x24 km2 size. Clarify.*

Agreed, it is the average of the full-resolution iDOAS measurements (from the nadir iDOAS pixel only) over the length of flight track within the OMI pixel. The captions and text have been amended to use the term "line-average".

**Figure 7 and similar:** *Specify colors for elevation/albedo subplot.*

A description of the colours has been added to the caption.

**Figure 8:** *Specify which are OMI and which are SCIAMACHY observations (maybe in legend?)*

Yes, specifying which are OMI and SCIAMACHY is a good idea. This has been done.

---

## Author Comment (AC2) · 20 Jun 2017

**AMT-2016-366: Responses to Anonymous Reviewer 3**

Stephen Broccardo

June 20, 2017

Thank you for your review and comments, which have added substantial value to the paper.

**Reviewer comment 1:** *Four research flights are analysed and discussed in quite some detail, but the interpretation of differences between airborne and satellite retrievals could go more into depth. Based on the present manuscript, the reader might get the impression that systematic differences in tropospheric $NO_2$ columns between satellite and iDOAS can be explained solely (or largely) by horizontal variability in the tropospheric $NO_2$ columns on a scale that is smaller than the typical size of satellite pixels. What would be particularly relevant is to investigate further the potential impact of profile shape assumptions for $NO_2$ and aerosols in explaining the difference between satellite and airborne measurements over the most polluted regions. Close to major point sources one may expect not only to find locally quite extreme tropospheric $NO_2$ column abundances, but at the same locations also the $NO_2$ profile shape may deviate considerably from other places further away from the main sources. In this context, it may be relevant to distinguish explicitly four profiles: the true profile at the spatial resolution of the aircraft measurements (P true air), the profile used in the airborne retrieval (P prior air), the true profile at the resolution of the satellite measurements (P true sat) and the profile used in the satellite retrieval (P prior sat). Differences in tropospheric $NO_2$ column retrievals (space-borne vs airborne) cannot be interpreted without taking into account these four profile shapes in the discussion. How much do the authors think P rue air can deviate from P prior air close to the main sources (same for P true sat and P prior sat). Furthermore the AMF is not only affected by the (different) profile shapes, but also by the block-AMFs, and these are not identical for the satellite and the airborne point of view. This should be taken into account as well.*

*Despite the length of this comment, I would suggest to add just one or two paragraphs addresssing this point and providing some first order estimates. It would for instance be enlightening to the reader if the impact of making*

*wrong profile shape assumptions is worked out for one hypothetical scenario. For instance (it is up to the authors to deviate from this concrete suggestion): scale height for P true air is 0.2km (e.g. close to strong isolated source); scale height for P true sat is 0.4km (averaged over a larger region the true profile is less dominated by the local source); scale heigh for P prior air is 0.6km (this number is used in the present study); scale height for P prior sat is taken from the profile used in DOMINOv2 product over this region. Block AMFs should be applied for a representative SZA and surface reflectance. When combined, this information should provide the reader with a first order quantitative estimate of local AMF fluctuations near a strong plume: to what extent can this explain the discrepancy between the satellite and airborne retrieval? Or perhaps it is concluded that - when taking this effect into account - the observed discrepancy increases even further.*

**Response**

This is a good idea. We have developed a further suite of vertical profile scenarios, based on Scenarios 11 and 12. These scenarios have a variety of scale-heights from 0.2km to 1.4km.

Like in the previously presented model scenarios, AMFs were calculated for each profile shape with all the permutations of SSA set at 0.82, 0.90, and 0.98; and surface albedo set at 0.02, 0.05, 0.08, and 0.11. All these profiles, like their parent profiles of scenario 11 and 12, have the surface elevation set to 1400m. In order to address the reviewer's next comment pertaining to the use of a fixed AOT, calculations were repeated with the AOT set to 0.1, 0.3 and 0.5.

What is remarkable from Figure 1 in this response, is that the trend in AMF with decreasing scale-height is negative for scenarios without an elevated layer, and positive for scenarios with such a layer. Such layers have been observed in this and other measurement campaigns. This result implies that close to a surface $NO_2$ source, such as the city of Johannesburg, the error from incorrect choice of *a-priori* vertical $NO_2$ profile cannot be determined without an actual profile measurement.

**Action**

The following paragraphs and the figure have been added to the discussion on page 14:

It is instructive to evaluate the potential air-mass factor error that might be made by assuming an incorrect vertical profile of $NO_2$. Several more radiative-transfer modelling scenarios are introduced, based on scenarios 11 and 12, i.e. with an exponentially-decreasing profile, surface elevation set at 1400m, some profiles with an elevated layer of $NO_2$ and some without. The scale height of the profiles is varied from 1400m to 200m, and radiative

[Figure]

Figure 1: AMFs for scenarios of varying scale-height, for aircraft viewing geometry (left) and satellite viewing geometry (right). For each scenario of scale-height and AOT, variability in the AMF is due to variations in surface albedo and single-scattering albedo.

transfer calculations are done at a representative solar zenith angle of 55°. Once again air-mass factors for permutations of AOT of 0.1, 0.3, and 0.5, and SSA of 0.82, 0.90, and 0.98 are calculated. Results for aircraft- and satellite viewing geometry are presented in Fig. 8. It can be seen that the AMF increases for scenarios with an elevated $NO_2$ layer as the vertical profile scale-height is decreased. In contrast, the AMF for scenarios without such a layer decreases as the scale-height is reduced. In the satellite viewing geometry, the behaviour is slightly different, with a flattening off of the AMFs with scale-heights of 600m and 400m, compared to the aircraft viewing geometry. This behaviour can likely be explained by examination of the block-AMFs for the two cases, however such analysis is beyond the scope of the present study.

We might estimate the VCD error arising from AMF uncertainty for the iDOAS using two profiles: the true profile at the spatial scale of the instrument, $P_{true}$ and the profile used in the AMF calculation $P_{prior}$, along with the associated AMFs: $AMF_{true}$ and $AMF_{prior}$. If $P_{prior}$ is an exponentially-decreasing profile with scale-height of 1000m either with- or without an elevated layer, $AMF_{prior}$ will lie between approximately 1.6 and 2.6. Close to a surface source of $NO_2$ $P_{true}$ might have a much smaller scale-height, for example 400m. In the case of a profile with an elevated layer, $AMF_{true}$ should be between 2.5 and 3.2. Using the mid-points of the uncertainty ranges of $AMF_{true}$ and $AMF_{prior}$, this will lead to a 26% overestimation of the VCD. In the case of $P_{true}$ having no elevated layer, $AMF_{true}$ will lie between approximately 1.2 and 2.3, leading to a 20% underestimation of the

VCD from the use of $\text{AMF}_{prior}$.

In the case of a satellite measurement, a representative profile for the satellite pixel is likely to have a larger scale-height, since more background areas will be included in the measurement along with the surface source and the discrepancy between $\text{AMF}_{prior}$ and $\text{AMF}_{true}$ will be less, but will behave in a similar manner to that described above. This highlights the need for an improved $\text{P}_{prior}$ as the spatial resolution of the measurement improves.

***Reviewer comment 2:*** *Although aerosols are not entirely neglected in this study, they receive little attention considering the fact that for all four flights - each covering distances of hundreds of kilometers - just one fixed value is assumed for the AOT. It is quite remarkable that the uncertainty range of the AMF is derived using a look-up table that does include variability of the single-scattering albedo, but not of the AOT. Over a region where the variability in $NO_2$ is so large, it is almost unthinkable that the AOT can be approximated with a single value. To some extent the same argumentation as given above (in the vicinity of a strong pollution source the $NO_2$ profile shapes may show considerable spatial variability) can be given here as well: in the same region the AOT may show a substantial variability (although probably less extreme than for $NO_2$). In my opinion this point should at least be mentioned. It would be even better to find satellite AOT data (e.g. from MODIS) for the days of the research flights to provide more insight into the relevant parameter*

Agreed. The approach taken to constrain AMF uncertainty arising from profile shape, SSA, surface albedo, surface elevation and SZA in this paper, by calculating all the permutations of these parameters, is extended to the AOT. The above parameters are further permuted with AOT's of 0.1, 0.3 and 0.5. 2-dimensional plots of AMF vs SZA (which could be thought of as slices of the discussion paper's Figure 5) are shown below for each surface albedo, with the original modelling highlighted in orange, and the additional permutations with the and lower and higher AOT in grey and blue-green respectively.

The increase in the range of AMF uncertainty derived from the present approach of modelling all permutations, as a result of the extra two AOT's used is not as large as might be anticipated. Nevertheless, the new values of minimum and maximum AMF will be used and figures, tables and discussion in the manuscript will be updated. In addition, a mistake in scenario 12, where the incorrect vertical profile of $NO_2$ was used, has been corrected.

***Reviewer comment 3:*** *In the manuscript the discrepancies found between iDOAS and OMI (SCIAMACHY) are not compared to results from other validation studies, e.g. where OMI retrievals are compared to MAX-DOAS observations. In the last years many of such studies were doen, with MAX-*

[Figure]

Figure 2: AMF versus SZA plots at different surface albedo's for scenarios with the surface elevation at sea level. AOT=0.3 (as in the discussion manuscript) is plotted in orange, and the additional scenarios of AOT=0.1 and 0.5 are plotted in grey.

*DOAS instruments either in rural or in urban regions. It would be valuable to link the findings of this study to findings in such inter-comparisons.*

**Response and Action**

The following paragraph has been added:

Comparison studies of ground-based multi-axis DOAS (MAX-DOAS) instruments with satellite measurements have given mixed results. Some studies (Irie et al, 2008; Hains et al 2010) showing MAX-DOAS results consistently lower than OMI. Kanaya et al (2014) shows DOMINOv2 biases of up to 50% lower than the MAX-DOAS, although the bias improves when only

[Figure]

Figure 3: AMF versus SZA plots at different surface albedo's for scenarios with the surface elevation at 1400m above sea level. AOT=0.3 (as in the discussion manuscript) is plotted in orange, and the additional scenarios of AOT=0.1 and 0.5 are plotted in grey and blue-green respectively.

remote surface sites are considered. This is attributed to both horizontal inhomogeneity within the OMI pixels and the inability of OMI to observe $NO_2$ close to the surface.

**Reviewer comment 4:** *On section 2: please provide some more details on the iDOAS observations. For instance: the field of view, number of pixels in across-track direction.*

**Reviewer comment 5:** *I am missing a formula that describes how VCD's are derived precisely from the (differential) slant column measurements. In my opinion, this should be described in more detail, although it has already been described elsewhere in full detail.*

**Response**

These two comments are similar to comments made by Reviewer 1. Details of the iDOAS and the retrieval have been expanded.

***Reviewer comment 6:*** *The statements in Sect 3 are quite general. The words "usually" (p.3,l.3) and "frequently" (p.6,l.8) suggest a large number of profiles that are measured. However, these are not shown. Furthermore it is not clear if the profiles that are measured are representative for the plume or for more remote regions (see also the first comment).*

**Response**

It is true that these words express more confidence than what is warranted by the limited number of profiles measured during this campaign. The confidence that the authors feel is not from the profiles measured at the start and end of each iDOAS-measurement flight leg, which frequently were in background conditions; but rather from the literature on stable discontinuities over the Highveld, which is based on an analysis of long-term observations, and on how in the literature, elevated trace-gas and aerosol layers are frequently associated with these stable discontinuities, an observation corroborated by our own measurements. The impact of the presence or absence of such layers on the AMF has emerged as a finding of the present study, discussed above in the response to the reviewer's first comment.

**Action**

The language has been changed to reflect the above discussion.

***Reviewer comment 7*** *Figure 5 could be better readable if a grid was plotted on the left and right side of each cube. Furthermore it could be beneficial to use colours instead of different line styles and to provide a legend.*

**Response**

This was attempted, however the increased clutter in the diagram made it even more difficult to read.

**Action**

The 3-d figure will be replaced with the conventional plots shown in response to point 2 above. A 3-d figure will be retained to illustrate the principle of a minimum- and maximum-AMF plane.

***Reviewer comment 8:*** *P.15, l24-25: "... indicating that ... 9 Aug". The terminology 'aged' versus 'young' might cause confusion, as some readers might wrongly think of 'photochemical aging'. It might be that what is here called an 'aged plume' is actually a region where the $NO_2$ profile is less shallow than for a 'young plume', and more in line with the prior $NO_2$ profile shape used for the OMI and/or iDOAS retrievals (see also the first*

*comment). If this is the case, then one cannot say that OMI would be limited in its ability to capture the higher NO₂ gradient in the young plume because it is 'young'; for instance it could be more appropriate to say that the AMF derived using the prior profile shape used in the DOMINO product better matches the profile shape of an aged plume than the profile of a young plume. Please comment on this.*

**Response**

Indeed, the terminology may cause confusion. What is implied by an "aged" plume is one that is more dispersed in both the vertical and horizontal directions. The shallower horizontal gradients in a more dispersed, "aged", plume are one reason why OMI might be better able to observe a more representative VCD, since the horizontal distribution of $NO_2$ across the OMI pixel is more homogeneous. This is what was meant in the discussion paper.

As the reviewer points out, the vertical dispersion of $NO_2$ in an "aged" plume could mean that the actual profile shape is closer to the *a-priori* profile used to calculate the satellite AMF. In addition, as shown in the figure above, potential errors in the AMF are less for profiles with a larger scale-height, and the divergence in the sign of the error for scenarios with- and without elevated layer found at low scale-heights disappears. This implies that AMF uncertainty will be smaller further downwind.

**Action**

The above discussion has been added, and the terminology has been changed.